# Orthogonal Concept Erasure for Diffusion Models

**Yuhao Sun** [1]   **Lingyun Yu** [1]   **Haoxiang Xu** [1]   **Fengyuan Miao** [1]   **Zhuoer Xu** [2]   **Hongtao Xie** [1]

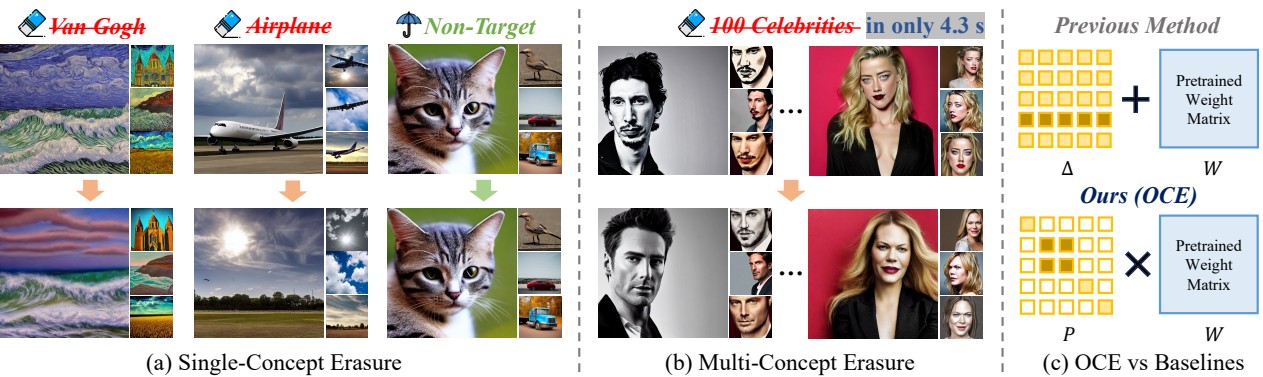

*Figure 1.* Our proposed method, OCE, achieves strong performance in both single-concept and multi-concept erasure. (a) OCE effectively removes target concepts in object and artistic style erasure while preserving non-target concepts. (b) OCE supports efficient large-scale multi-concept erasure of up to 100 concepts at once, requiring only 4.3 s. (c) OCE reformulates previous additive editing as multiplicative orthogonal editing, introducing a new editing-based concept erasure paradigm.

## Abstract

Concept erasure has emerged as a promising approach to mitigate undesired or unsafe content in diffusion models, yet existing methods still face significant limitations. While training-based methods are effective, their high computational cost limits scalability. Editing-based methods are more efficient and deployment-friendly, yet they struggle to simultaneously achieve precise concept erasure and preserve overall generative capacity. We identify this core limitation of the editing-based methods as reliance on additive parameter updates. Our empirical analysis reveals that concept semantics primarily depend on *neuron direction* rather than *neuron magnitude*, while overall generative capacity relies on the *angular geometry* of neurons. As additive updates inherently entangle direction, magnitude, and angular geometry, they inevitably introduce unintended interference between concept erasure and overall generation performance. To address this, we propose **Orthogonal Concept Erasure (OCE)**, which reformulates editing-based erasure as multiplicative parameter updates from a geometric perspective. Specifically, OCE applies layer-wise orthogonal transformations derived from a closed-form solution to the parameters, enabling precise concept erasure while preserving the neuron magnitude and angular geometry. Furthermore, to address conflicting constraints in multi-concept erasure, OCE introduces a subspace-level objective with structured subspace manipulation, yielding a more effective and scalable erasure. Extensive experiments on single- and multi-concept erasure demonstrate that OCE outperforms existing methods in concept erasure and non-target preservation, erasing up to 100 concepts in 4.3 s. Code: https://github.com/HansSunY/OCE.

## 1. Introduction

Text-to-image (T2I) diffusion models (Ho et al., 2020; Song et al., 2020a;b; Rombach et al., 2022; Ho & Salimans, 2022; Nichol et al., 2021; Ramesh et al., 2022) have achieved remarkable performance in synthesizing high-fidelity and diverse images from text prompts. However, their impressive generative capacity can lead to the generation of undesirable concepts, including copyrighted content (Jiang et al., 2023), offensive or sensitive visual attributes (Schramowski et al.,

---

[1]University of Science and Technology of China [2]Ant Group. Correspondence to: Lingyun Yu <yuly@ustc.edu.cn>.

*Proceedings of the 43rd International Conference on Machine Learning*, Seoul, South Korea. PMLR 306, 2026. Copyright 2026 by the author(s).

2023; Zhang et al., 2024c), and identity-related information (Carlini et al., 2023; Mirsky & Lee, 2021), raising concerns about safety, ethics, and privacy. Consequently, concept erasure has emerged as a critical research direction.

The task of concept erasure involves precisely removing the specific concepts from the pretrained model while preserving the prior concepts. Existing concept erasure techniques can be broadly categorized into inference-time intervention, training-based and editing-based methods. Inference-time intervention methods (Schramowski et al., 2023; Yoon et al., 2024; Wang et al., 2025) adjust sampling trajectories without modifying model parameters, but they are vulnerable to bypassing. Training-based methods (Kumari et al., 2023; Gandikota et al., 2023; Lyu et al., 2024; Lu et al., 2024) achieve concept erasure by fine-tuning a subset of model parameters with carefully designed objectives to remove target concepts. While effective, these approaches typically require multiple rounds of optimization, leading to substantial computational time and overhead, which significantly limits their practicality in real-world scenarios. By contrast, editing-based methods (Gandikota et al., 2024; Gong et al., 2024; Li et al., 2025c) operate directly on model parameters, such as the projection weights in cross-attention layers, using closed-form solutions. This makes editing-based methods more efficient and scalable to real-world scenarios like multi-concept erasure. However, despite their efficiency, existing editing-based methods still suffer from several limitations. In particular, they often exhibit insufficient erasure precision, struggle to reliably preserve prior concepts, and often rely on relatively complex erasure pipelines, which makes them less concise and principled than desired.

In this work, we argue that the core limitation of existing editing-based methods (Gandikota et al., 2024; Gong et al., 2024; Li et al., 2025c) lies in how concept erasure is formulated. Most existing methods define concept erasure as additive updates of model parameters $W + \Delta$, as shown in Fig. 1(c). While simple and flexible, such additive updates simultaneously perturb neurons in both magnitude and direction. More critically, even small additive changes can arbitrarily alter the angular geometry of neurons, including inter-neuron angles and correlations. Inspired by previous studies on hyperspherical energy (Liu et al., 2017; 2018; Chen et al., 2020; Qiu et al., 2023), we design a toy experiment, which reveals that (1) *neuron direction* is crucial for encoding concept semantics, whereas magnitude has little effect, and (2) preserving the *angular geometry* is essential for maintaining the model's overall generative capabilities. Consequently, additive updates inevitably entangle direction, magnitude, and angular geometry, resulting in unstable erasure and poor preservation of prior concepts. **This motivates a geometric perspective on concept erasure: rather than performing unconstrained additive corrections, effective erasure should directly manipu-** late neuron directions, which encode concept semantics, while preserving their magnitudes and intrinsic angular geometry.

From this geometric perspective, we propose **O**rthogonal **C**oncept **E**rasure (OCE) for diffusion model. Instead of additive updates, OCE applies a layer-wise orthogonal transformation derived from a closed-form solution to model parameters, precisely rotating neuron directions for concept erasure while preserving the neuron magnitude and angular geometry. Furthermore, to address conflicting constraints in multi-concept erasure, we extend the vector-wise erasure objective to a subspace-level projection objective. Specifically, we represent the target and anchor concepts as their respective subspaces and then minimize the components of the target subspace that lie outside the orthogonal complement of the anchor subspace, which provides a more structured and gentle form of concept erasure. By explicitly controlling neuron directions through a multiplicative and structured update, OCE achieves precise erasure of target concepts while preserving the generation ability of non-target concepts. It provides a principled, and efficient closed-form solution for both single- and multi-concept erasure scenarios. The main contributions of this work are summarized as follows:

- We propose OCE, a geometry-driven, editing-based method that performs concept erasure via layer-wise orthogonal transformations, enabling precise removal of target concepts while preserving the intrinsic angular geometry of pretrained models.

- We introduce a structured subspace-level projection objective with closed-form solutions. By extending vector-wise erasure to a subspace projection formulation, OCE provides a principled, scalable, and efficient solution for both single- and multi-concept erasure, achieving improved erasure precision and prior concept preservation.

- Extensive experiments across diverse erasure tasks, demonstrating that OCE consistently outperforms existing training-based and editing-based methods in terms of erasure effectiveness and overall generation quality.

## 2. Related Work

### 2.1. Concept Erasure in Diffusion Models

Text-to-image diffusion models can generate copyrighted, offensive and privacy content, raising serious concerns for real-world deployment. To better regulate the content generated by these models, early approaches focused on retraining with curated datasets (Rombach, 2022) or introducing output filters (Rando et al., 2022) to block undesirable generations. However, retraining is computationally expensive, while output filters can be easily bypassed. This has

led to increasing interest in concept erasure, which selectively removes target concepts from models, categorized into three paradigms. Inference-time intervention methods (Schramowski et al., 2023; Yoon et al., 2024; Jain et al., 2024; Wang et al., 2025) modify sampling trajectories or guidance signals during generation to suppress undesired concepts. Training-based methods (Kumari et al., 2023; Gandikota et al., 2023; Zhang et al., 2024a; Lyu et al., 2024; Lu et al., 2024; Zhang et al., 2024b; Srivatsan et al., 2025; Nguyen et al., 2025; Sun et al., 2025) fine-tune model parameters with specific erasure objectives and regularization terms to preserve general generation quality. Editing-based method (Orgad et al., 2023; Gandikota et al., 2024; Gong et al., 2024; Li et al., 2025c; Lin et al., 2025) directly modify model parameters with closed-form updates for more efficient concept removal. In contrast to existing additive update paradigms in editing-based approaches, our method reformulates the concept erasure paradigm as an orthogonal transformation from a geometric perspective, enabling more structured and effective concept erasure.

## 2.2. Orthogonalization in Parameter Space

Orthogonalization has been widely explored in parameter space as a principled way to fine-tuning models and reducing interference across objectives. In continual and multi-task learning, orthogonal gradient methods (Bennani et al., 2020; Farajtabar et al., 2020) project gradients onto the orthogonal complement of past task subspaces or decompose updates to mitigate negative transfer between tasks. More recently, orthogonal transformations have been introduced for parameter-efficient fine-tuning. Orthogonal Fine-Tuning (OFT) (Qiu et al., 2023) and its variants (Liu et al., 2023; Ma et al., 2024; Qiu et al., 2025) apply structured orthogonal transformations to weight matrices, preserving the intrinsic pretrained structure. Despite their success, existing parameter-space orthogonalization methods primarily focus on stabilizing training dynamics or text-to-image customization. In contrast, our work leverages orthogonal transform as a geometric tool for closed-form concept erasure, enabling targeted concept suppression through a structured transformation in parameter space, while maintaining overall generation capacity of the model.

## 3. Geometric Analysis of Concept Erasure

In this section, we analyze concept erasure from a geometric perspective and argue that its effectiveness and stability are fundamentally governed by the angular structure of the neuron. Our analysis is centered around two key claims: **(C1)** Concept semantics in diffusion models are more strongly associated with neuron directions than with magnitudes. **(C2)** Preserving inter-neuron angular geometry is critical for maintaining the overall generation quality.

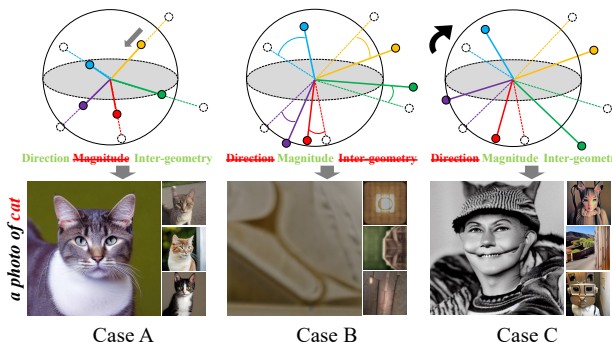

*Figure 2.* A toy experiment to demonstrate the importance of the angular information for concept expression in diffusion models.

### 3.1. Concept Expression in Neuron Angular Geometry

We analyze controlled geometric transformations of the cross-attention projection matrices in diffusion models to disentangle the roles of magnitude, direction, and inter-neuron geometry in concept expression. Specifically, we focus on the key and value matrices, denoted as $W$, which play a central role in mapping text embeddings to visual semantics. We design three controlled modification of $W$:

**Case A: Magnitude-only scaling.** We scale the weights by a scalar factor $\alpha$ as

$$\tilde{W} = \alpha W, \quad \alpha \in (0, 1), \tag{1}$$

which changes neuron magnitudes only.

**Case B: Neuron-wise orthogonal rotation.** We apply independent orthogonal transformations to individual neurons:

$$\tilde{w}_i = Q_i w_i, \quad Q_i^\top Q_i = I, \tag{2}$$

where $w_i$ denotes the $i$-th column of $W$ and each $Q_i$ is a neuron-wise orthogonal matrix. This preserves neuron magnitudes but breaks the directions and inter-neuron angular geometry.

**Case C: Layer-wise orthogonal rotation.** We apply a shared orthogonal transformation $Q$ to the entire layer:

$$\tilde{W} = QW, \quad Q^\top Q = I, \tag{3}$$

which changes neuron directions only.

We evaluate the effects of these three modifications on the generation of the concept "cat" as shown in Fig. 2. For Case A, scaling the weights by $\alpha = 0.5$ has negligible effect on the generation of concept "cat". By contrast, applying a small layer-wise orthogonal rotation (Case C) results in a clear semantic shift of the "cat" concept in the generated images, confirming that neuron directions are the primary carriers of concept semantics (**C1**). In addition, applying independent neuron-wise rotations (Case B) substantially degrades overall image quality, highlighting that maintaining the relative angular geometry among neurons is crucial

for generation capability (**C2**). These observations are also consistent with prior studies on hyperspherical energy (Liu et al., 2017; 2018; Chen et al., 2020; Qiu et al., 2023).

### 3.2. Additive vs. Orthogonal Updates

Based on the above discussion, we analyze why layer-wise orthogonal update is a principled choice for concept erasure.

**Additive updates.** An additive update takes the form

$$W^* = W + \Delta, \quad w_i^* = w_i + \delta_i, \tag{4}$$

for each neuron $w_i$. These updates primarily rotate neuron directions $cos\theta_i^* \neq \cos\theta_i$, but also alter neuron magnitudes $\|w_i^*\| \neq \|w_i\|$ and inter-neuron angles $\cos\phi_{ij}^* \neq \cos\phi_{ij}$. These combined effects explain why additive updates often fail to precisely erase target concepts and why they can disrupt non-target concepts by disturbing the angular geometry that encodes semantic structure. As a result, additive updates often fail to precisely remove target concepts, since magnitude changes contribute little to semantic erasure, and they also degrade the preservation of non-target concepts due to changes in angular geometry.

**Layer-wise orthogonal updates.** A layer-wise orthogonal update takes the form

$$W^* = QW, \quad Q^\top Q = I, \quad w_i^* = Qw_i.$$

Such a multiplicative update rotates all neurons in the layer by a shared orthogonal transformation. As a result, neuron magnitudes are preserved, $\|w_i^*\| = \|w_i\|$, and pairwise inter-neuron angles remain unchanged, $\cos\phi_{ij}^* = \cos\phi_{ij}$. The update modifies directions in a structured manner, without introducing changes to magnitude or angular geometry.

In summary, additive updates entangle changes in direction, magnitude, and angular geometry, making precise erasure and preservation of non-target concepts difficult. In contrast, layer-wise orthogonal updates modify neuron directions while strictly preserving magnitudes and angular geometry, which better matches the geometric structure underlying semantic representations. This makes orthogonal transformations a more principled choice for concept erasure.

## 4. Problem Formulation

In text-to-image diffusion models, semantic concepts are encoded through text embeddings $c$ processed by a pretrained CLIP text encoder. During image generation, these embeddings are injected into the denoising U-Net with cross-attention layers, where the key and value projection matrices, denoted as $W_k$ and $W_v$, play a central role in shaping semantic alignment. For simplicity, we denote a generic projection matrix as $W$.

### 4.1. Additive Closed-Form Concept Erasure

Editing-based concept erasure aims to modify the projection matrix $W$ such that target concepts are suppressed while non-target concepts remain unaffected. Following prior approaches, we consider three sets of concept embeddings: an erasure set $\mathcal{E} = \{c_1^{(i)}\}_{i=1}^{N_E}$ containing target concepts to be removed, a retain set $\mathcal{R} = \{c_0^{(j)}\}_{j=1}^{N_R}$ containing non-target concepts to be preserved, and an anchor set $\mathcal{A} = \{c_*^{(i)}\}_{i=1}^{N_E}$, where each target concept $c_1^{(i)}$ is mapped onto an anchor concept $c_*^{(i)}$ that serves as a semantically similar surrogate. We stack the embeddings of the erasure, anchor and retain sets into matrices $C_1$, $C_*$ and $C_0$ respectively. Additive erasure methods, with UCE as a representative example, optimize a parameter perturbation $\Delta$ on $W$ through the following least-squares formulation:

$$\min_\Delta \|(W + \Delta)C_1 - WC_*\|_F^2 + \|(W + \Delta)C_0 - WC_0\|_F^2. \tag{5}$$

The first term enforces effective erasure by mapping each target concept to its anchor, while the second term is designed to preserve non-target concept. This objective provides a closed-form solution, enabling efficient parameter updates without iterative optimization. Despite its simplicity and efficiency, as shown in Sec. 3, additive erasure suffers from intrinsic geometric limitations, which constrain both erasure precision and the preservation of non-target concepts.

### 4.2. From Additive to Orthogonal Erasure

Motivated by the geometric analysis in Sec. 3, we reformulate concept erasure from additive parameter perturbations to multiplicative orthogonal transformations. We seek an orthogonal matrix $P$ that modifies the projection space in a structured manner. This leads to the following simple orthogonal erasure objective:

$$\min_{P^\top P = I} \|PWC_1 - WC_*\|_F^2 + \|PWC_0 - WC_0\|_F^2. \tag{6}$$

Compared to additive formulations, this orthogonal perspective provides a geometrically principled foundation for concept erasure, aligning with the observations in Sec. 3 that concept semantics are direction-sensitive and that preserving angular geometry is critical for maintaining generation quality. The solution and further refinement of this objective are presented in the method section.

## 5. Orthogonal Concept Erasure (OCE)

### 5.1. Closed-Form Solution for Orthogonal Update

**Solving the Vector-wise Objective.** After defining the initial vector-wise objective for orthogonal erasure in Eq. 6, we first discuss how to solve this problem in closed form.

First, we stack the two terms in Eq. 6 into block matrices:

$$A = [WC_1, \ WC_0], \quad B = [WC_*, \ WC_0]. \quad (7)$$

The objective is equivalently written as

$$\min_{P^\top P = I} \|PA - B\|_F^2. \quad (8)$$

This is equivalent to the following trace maximization:

$$\max_{P^\top P = I} \operatorname{tr}(PM), \qquad M = BA^\top. \quad (9)$$

Detailed derivation is shown in Appendix. A.1. The closed-form solution follows from the classical Procrustes theorem (Schönemann, 1966). Specifically, we compute the cross-covariance matrix $M$ and perform its Singular Value Decomposition (SVD):

$$M = U\Sigma V^\top, \quad (10)$$

from which the optimal orthogonal transformation is

$$P = UV^\top. \quad (11)$$

**Construction of $M$.** When calculating $M$, we expand it in terms of the original blocks gives

$$M = (WC_*)(WC_1)^\top + (WC_0)(WC_0)^\top \quad (12)$$
$$= W(C_*C_1^\top + C_0C_0^\top)W^\top. \quad (13)$$

For the term involving $C_0C_0^\top$, we further decompose the retain set into two subsets: a set of generic concepts $\mathcal{R}_g$ that should be universally preserved, and a set of local neighboring concepts $\mathcal{R}_n$ that are specific to the current erasure task. Correspondingly, we write $C_0 = [\,C_g, \ C_n\,]$, where $C_g$ and $C_n$ stack the embeddings from $\mathcal{R}_g$ and $\mathcal{R}_n$, respectively. The cross-covariance term then becomes

$$W(C_0C_0^\top)W^\top = W(C_gC_g^\top + C_nC_n^\top)W^\top. \quad (14)$$

The generic preservation term $C_gC_g^\top$ is independent of the target concept and can therefore be precomputed once and reused across different erasure tasks. Concretely, we compute all the token embeddings $c$ over the COCO-30k dataset and precompute

$$K_0 = C_gC_g^\top = \mathbb{E}_c[cc^\top], \quad (15)$$

which serves as a global preservation prior shared by all erasure instances. Finally, $M$ can be written as

$$M = W(C_*C_1^\top + K_0 + C_nC_n^\top)W^\top. \quad (16)$$

This objective achieves strict vector-wise alignment between target embeddings and anchors for erasure and preservation of the generation capacity. It is effective for single-concept erasure. However, in multi-concept erasure scenario, simultaneously enforcing exact alignment for multiple targets introduces conflicting constraints, which can interfere with each other and degrade erasure performance. This motivates a more global, subspace-level erasure objective.

## 5.2. Subspace-level Erasure Objective

Instead of enforcing vector-wise alignment between target and anchor embeddings, we further reformulate concept erasure as a geometric subspace suppression problem, where erase directions are discouraged from lying in the orthogonal complement of the anchor subspace. This formulation better respects the intrinsic geometry of the latent space and enables a more structured erasure.

**Target and Anchor Subspaces.** We first define the target and anchor subspaces in the transformed feature space induced by $W$. Given sets of concept embeddings $C_1$ and $C_*$, we first map each embedding through $W$ and normalize the resulting vectors. Then, an orthonormal basis for each subspace is obtained via Gram–Schmidt orthogonalization. Formally, we denote $G = \operatorname{orth}(WC_1)$ and $G_* = \operatorname{orth}(WC_*)$

where $\operatorname{orth}(\cdot)$ returns an orthonormal basis spanning the column space of its argument. The corresponding orthogonal projection matrices are given by

$$R = GG^\top, \qquad R_* = G_*G_*^\top. \quad (17)$$

**Subspace Suppression.** Instead of enforcing point-wise alignment between $WC_1$ and $WC_*$, we consider a subspace-level suppression objective that discourages the transformed target subspace from collapsing into the orthogonal complement of the anchor subspace. Formally, we seek an orthogonal transformation $P$ that pushes the target subspace away from directions orthogonal to the anchor space, while preserving the retain set $C_0$ through vector-wise constraints:

$$\min_{P^\top P = I} -\|PR - R_{*,\perp}\|_F^2 + \|PWC_0 - WC_0\|_F^2, \quad (18)$$

where $R_{*,\perp} = I - R_*$ is the projector onto the orthogonal complement of the anchor subspace.

This objective can then be equivalently rewritten as a trace maximization:

$$\max_{P^\top P = I} \operatorname{tr}(PM_{\text{total}}), \quad (19)$$

where $M_{\text{total}}$ combines contributions from target suppression and local retain:

$$M_{\text{total}} = -R(I - R_*) + W(K_0 + C_nC_n^\top)W^\top. \quad (20)$$

Detailed derivation is shown in Appendix. A.2. Here, we still write $C_0 = [C_g, C_n]$ as Eq. 14 and incorporate the global preservation prior $K_0$ to enforce global retention.

Eq. 18 also corresponds to a classical orthogonal Procrustes problem. Following the same procedure as Eq. 10 and 11, we perform SVD on $M_{\text{total}}$ and compute the closed-form solution as $P = UV^\top$.

*Table 1.* Evaluation of Object Erasure on CIFAR-10. We report $Acc_e$ (lower indicates more successful erasure), $Acc_s$ (higher indicates better preservation of non-target concepts) and $H_o$ (higher indicates better overall performance) across the first five categories. All values are expressed as percentages (%). The original model's performance is included for reference.

| Concept | Airplane | | | Automobile | | | Bird | | | Cat | | | Deer | | | Average | | |
|---|---|---|---|---|---|---|---|---|---|---|---|---|---|---|---|---|---|---|
| | $Acc_e\downarrow$ | $Acc_s\uparrow$ | $H_o\uparrow$ | $Acc_e\downarrow$ | $Acc_s\uparrow$ | $H_o\uparrow$ | $Acc_e\downarrow$ | $Acc_s\uparrow$ | $H_o\uparrow$ | $Acc_e\downarrow$ | $Acc_s\uparrow$ | $H_o\uparrow$ | $Acc_e\downarrow$ | $Acc_s\uparrow$ | $H_o\uparrow$ | $Acc_e\downarrow$ | $Acc_s\uparrow$ | $H_o\uparrow$ |
| SD v1.4 | 96.06 | 98.92 | 7.58 | 95.75 | 98.95 | 8.15 | 99.72 | 98.51 | 0.56 | 98.93 | 98.60 | 2.12 | 99.87 | 98.49 | 0.26 | 98.07 | 98.69 | 3.79 |
| CA | 96.24 | 98.55 | 7.24 | 94.41 | 98.47 | 10.58 | 99.55 | 98.53 | 0.90 | 98.94 | 98.63 | 2.10 | 99.45 | 98.47 | 1.09 | 97.72 | 98.53 | 4.46 |
| ESD | 7.38 | 85.48 | 88.91 | 30.29 | 91.02 | 78.95 | 13.17 | 86.17 | 86.50 | 11.77 | 91.45 | 89.81 | 18.14 | 73.81 | 77.63 | 16.15 | 85.59 | 84.71 |
| FMN | 96.76 | 98.32 | 6.27 | 95.08 | 96.86 | 9.36 | 99.46 | 98.13 | 1.07 | 94.89 | 97.97 | 9.71 | 98.95 | 94.13 | 2.08 | 97.03 | 97.08 | 5.77 |
| UCE | 40.32 | 98.79 | 74.41 | 4.73 | 99.02 | **97.11** | 10.71 | 98.35 | 93.60 | 2.35 | 98.02 | 97.83 | 11.88 | 98.39 | 92.97 | 14.00 | 98.51 | 91.83 |
| MACE | 9.06 | 95.39 | 93.11 | 6.97 | 95.18 | 94.09 | 9.88 | 97.45 | 93.64 | 2.22 | 98.85 | 98.31 | 13.47 | 97.71 | 91.78 | 8.32 | 96.92 | 94.23 |
| RECE | 15.01 | 98.76 | 91.36 | 36.13 | 99.04 | 77.66 | 7.95 | 98.01 | 94.94 | 15.76 | 98.51 | 90.82 | 9.62 | 98.27 | 94.16 | 16.89 | 98.52 | 90.16 |
| SPEED | 57.78 | 98.74 | 59.15 | 10.33 | 98.52 | 93.89 | 38.48 | 98.23 | 75.66 | 0.36 | 98.57 | 99.10 | 2.17 | 98.31 | **98.07** | 21.82 | 98.47 | 87.16 |
| **Ours** | 6.89 | 98.85 | **95.89** | 8.79 | 99.07 | 94.98 | 3.40 | 98.30 | **97.44** | 0.15 | 98.71 | **99.28** | 3.81 | 98.45 | 97.31 | 4.61 | 98.68 | **97.01** |

*Table 2.* Evaluation of Artistic Style Erasure. We report CLIP Score (CS) for each style generation, where lower CS indicates better erasure. We also report FID and CS on MSCOCO-30k, where lower FID and higher CS indicate better preservation quality.

| Method | Van Gogh | | | Picasso | | | Monet | | |
|---|---|---|---|---|---|---|---|---|---|
| | CS↓ | MSCOCO | | CS↓ | MSCOCO | | CS↓ | MSCOCO | |
| | | FID↓ | CS↑ | | FID↓ | CS↑ | | FID↓ | CS↑ |
| CA | 25.01 | 8.11 | 26.26 | 26.33 | 7.40 | 26.28 | 24.20 | 7.02 | 26.25 |
| ESD | 20.10 | 13.60 | 26.29 | 21.10 | 12.20 | 25.82 | 19.73 | 13.51 | 25.87 |
| FMN | 22.36 | 13.85 | 26.06 | 21.24 | 14.63 | 26.00 | 25.23 | 14.25 | 26.09 |
| UCE | 21.22 | 14.53 | 26.45 | 22.84 | 7.65 | 26.24 | 23.17 | 14.44 | 25.77 |
| MACE | 23.49 | 12.28 | 26.21 | 23.77 | 12.08 | 26.24 | 21.70 | 12.28 | 26.25 |
| RECE | 22.78 | 8.05 | 26.49 | 24.01 | 7.67 | 26.22 | 22.30 | 8.65 | 25.79 |
| SPEED | 21.68 | 7.49 | 26.20 | 23.24 | 7.52 | 26.31 | 21.91 | 7.51 | 26.27 |
| **Ours** | 21.08 | 7.15 | 26.52 | 20.92 | 8.57 | 26.46 | 21.35 | 7.30 | 26.32 |

The proposed subspace-level formulation generalizes vector-wise erasure by lifting the objective from vector-wise alignment to subspace energy suppression. This design mitigates interference among multiple erased concepts and enables a unified closed-form solution, balancing targeted erasure with global representation stability.

Notably, we impose erasure at the subspace level while enforcing preservation in a vector-wise manner. This asymmetric design is intentional and principled. Compared with erasure, preservation requires finer-grained control. Vector-wise preservation ensures high-fidelity retention of individual non-target concepts and does not suffer from cross-concept interference. Using subspace-level constraints for preservation would be unnecessarily restrictive. Our ablation studies in Sec. 6.4 further confirm that this asymmetric formulation achieves a better balance between effective erasure and faithful preservation.

# 6. Experiments

In this section, we evaluate OCE on three representative concept erasure settings: single concept erasure, multi-concept erasure and implicit concept erasure (Appendix. 6.3). We compare OCE with several commonly adopted baselines, including CA (Kumari et al., 2023), ESD (Gandikota et al., 2023), FMN (Zhang et al., 2024a), UCE (Gandikota et al., 2024), MACE (Lu et al., 2024), RECE (Gong et al., 2024), SPEED (Li et al., 2025c). Our evaluation measures both *efficacy* and *specificity*: *efficacy* measures the target concept erasure, while *specificity* measures the preservation of unrelated concepts. All experiments are conducted on [1]SD v1.4. More implementation details can be found in Appendix. C.

## 6.1. Single Concept Erasure

**Evaluation Setup.** We evaluate single concept erasure on object erasure and artistic style erasure.

For **object erasure**, we edit ten models with each model targeted to erase one object class of the CIFAR-10 dataset. To evaluate *efficiency*, we generate 200 images using prompt "*a photo of the {erased class}*" with each model. All the images are classified using CLIP, where lower classification accuracy $Acc_e$ indicates more effective erasure. To assess *specificity*, we generate 200 images for each of the nine remaining classes using the prompt "a photo of the {*unrelated class*}" with each model, where higher classification accuracy $Acc_s$ indicates better preservation of unrelated concepts. The harmonic mean $H_o = \frac{2}{(1-Acc_e)^{-1}+(Acc_s)^{-1}}$ is adopted to assess the overall performance of erasure.

For **artistic style erasure**, we use CLIP Score (CS) and Fréchet Inception Distance (FID) as metrics. For *efficiency*, we generate 100 images using prompt "a painting in the style of {*style*}" for each style. We compute the average CS between the prompts and generated images, where lower CS indicates more effective erasure. For *specificity*, we generate images with the first 10K prompts from the MSCOCO-30k dataset, then compute CLIP scores and Fr'echet Inception Distance (FID) against the original images. Higher CLIP scores and lower FID indicate better preservation.

**Analysis and discussion.** Tab. 1 compares different methods on CIFAR-10. Our method achieves the best average

[1] https://github.com/CompVis/stable-diffusion

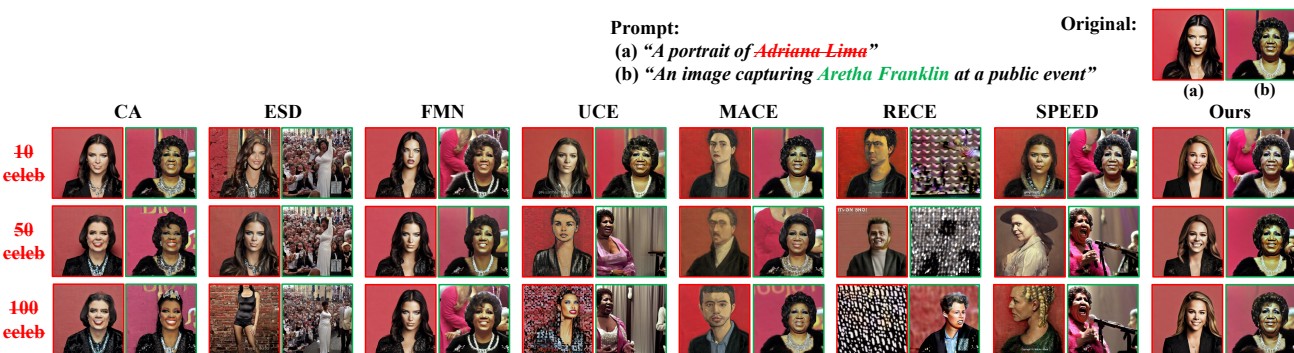

*Figure 3.* Quantitative comparison of the Multi-Concept Erasure in erasing celebrities. Red boxes indicate the erased target celebrity, while green boxes denote the preserved non-target celebrity. Our method achieves precise erasure of up to 100 concepts while effectively preserving non-target concepts. Notably, our method exhibits stable performance across varying erasure scales.

*Table 3.* Evaluation of Multi-Concept Erasure for celebrity with 10, 50, 100 target concepts. We report $Acc_e$, $Acc_r$, $H_o$ and Time (s). Our method consistently achieves strong erasure performance across different scales, effectively erasing up to 100 concepts while preserving non-target concepts with minimal degradation. It also demonstrates competitive efficiency, erasing 100 concepts in only 4.3 s on a single A100 GPU.

| Method | Erase 10 Celebrities | | | | MSCOCO | | Erase 50 Celebrities | | | | MSCOCO | | Erase 100 Celebrities | | | | MSCOCO | |
|---|---|---|---|---|---|---|---|---|---|---|---|---|---|---|---|---|---|---|
| | $Acc_e \downarrow$ | $Acc_s \uparrow$ | $H_o \uparrow$ | Time $\downarrow$ | CS $\uparrow$ | FID $\downarrow$ | $Acc_e \downarrow$ | $Acc_s \uparrow$ | $H_o \uparrow$ | Time $\downarrow$ | CS $\uparrow$ | FID $\downarrow$ | $Acc_e \downarrow$ | $Acc_s \uparrow$ | $H_o \uparrow$ | Time $\downarrow$ | CS $\uparrow$ | FID $\downarrow$ |
| SD v1.4 | 94.43 | 97.51 | 10.54 | – | 26.56 | – | 97.11 | 97.51 | 5.61 | – | 26.56 | – | 99.17 | 97.51 | 1.65 | – | 26.56 | – |
| CA | 25.52 | 87.29 | 80.38 | 1200 | 26.30 | 11.28 | 1.87 | 32.25 | 48.55 | 2640 | 25.66 | 16.97 | 0.80 | 11.13 | 20.01 | 5400 | 25.59 | 18.29 |
| ESD | 9.83 | 34.86 | 50.28 | 300 | 26.02 | 16.26 | 26.21 | 34.46 | 46.98 | 960 | 25.90 | 16.54 | 18.84 | 24.16 | 37.24 | 1800 | 25.97 | 17.24 |
| FMN | 44.90 | 66.43 | 60.24 | 12.0 | 26.36 | 11.84 | 35.82 | 56.10 | 59.87 | 28.0 | 26.37 | 11.66 | 51.48 | 49.20 | 48.86 | 53.0 | 26.40 | 11.43 |
| UCE | 2.45 | 90.89 | 94.10 | 1.5 | 25.83 | 17.37 | 38.90 | 88.31 | 72.23 | 1.8 | 26.19 | 19.86 | 58.67 | 76.79 | 53.74 | 2.1 | 25.23 | 28.07 |
| MACE | 1.86 | 96.72 | 97.42 | 200 | 26.31 | 13.87 | 3.82 | 93.23 | 94.68 | 840 | 25.44 | 17.68 | 8.02 | 91.60 | 91.79 | 1800 | 24.81 | 20.92 |
| RECE | 0.30 | 80.04 | 88.91 | 6.0 | 19.70 | 90.38 | 0.00 | 36.19 | 53.15 | 12.0 | 13.80 | 218.96 | 0.00 | 21.78 | 35.77 | 25.0 | 12.93 | 140.08 |
| SPEED | 3.22 | 96.52 | 96.65 | 3.8 | 26.55 | 10.96 | 4.00 | 95.07 | 95.53 | 4.2 | 26.30 | 16.21 | 5.21 | 92.67 | 93.72 | 5.0 | 26.29 | 18.40 |
| Ours | 0.61 | 96.92 | **98.14** | 3.2 | 26.53 | 11.20 | 2.66 | 95.91 | **96.62** | 3.6 | 26.43 | 16.03 | 3.44 | 94.42 | **95.48** | 4.3 | 26.33 | 18.33 |

performance across the five concept erasure tasks. Compared to the state-of-the-art method, our method reduces the average $Acc_e$ from 8.32 to 4.61, indicating substantially stronger erasure effectiveness. Meanwhile, it preserves $Acc_s$ almost perfectly, with only a 0.01% drop compared to the original model. As a result, our method achieves the highest average harmonic score $H_o$. The results for the remaining five classes are provided in Appendix. D.1.

Tab. 2 reports the results on artistic style erasure. Although ESD exhibits relatively strong erasure capability (lower CS), it significantly degrades the model's generation quality. RECE and SPEED preserve generation ability well; however, their erasure performance becomes unreliable in some cases. In contrast, our method consistently achieves effective erasure and preservation of overall generation quality.

### 6.2. Multi-Concept Erasure

**Evaluation Setup**   Multi-concept erasure poses a more challenging setting, as mass concepts are erased simultaneously. We follow the experiment setup in SPEED (Li et al., 2025c) for large-scale celebrity erasure, conducting experiments that erase 10, 50 and 100 celebrities, respectively.

In addtion, we collect a separate set of 100 celebrities as non-target concepts to evaluate preservation behavior. During evaluation, we employ five prompt templates for each celebrity. For both target and non-target sets, we generate 500 images in total by adjusting the number of images per celebrity accordingly. All generated images are then evaluated using the GIPHY Celebrity Detector (GCD) (Hasty et al., 2019), where we report the top-1 detection accuracy. $Acc_e$ measures the detection accuracy on erased target concepts and $Acc_s$ on non-target concepts. Meanwhile, to assess the overall performance of erasure, we compute the $H_o$ and report the CS and FID results on MSCOCO.

**Analysis and discussion.**   Tab. 3 further demonstrates the superiority of our method in multi-concept erasure. Compared to the state-of-the-art methods MACE and SPEED, our method achieves stronger erasure effectiveness while better preserving non-target concepts. In contrast, other methods tend to suffer from severe generation collapse as the number of erasure concepts increases. Notably, our method also offers clear efficiency advantages. While both SPEED and our approach are significantly faster than MACE, it is important to note that the reported runtime of SPEED does

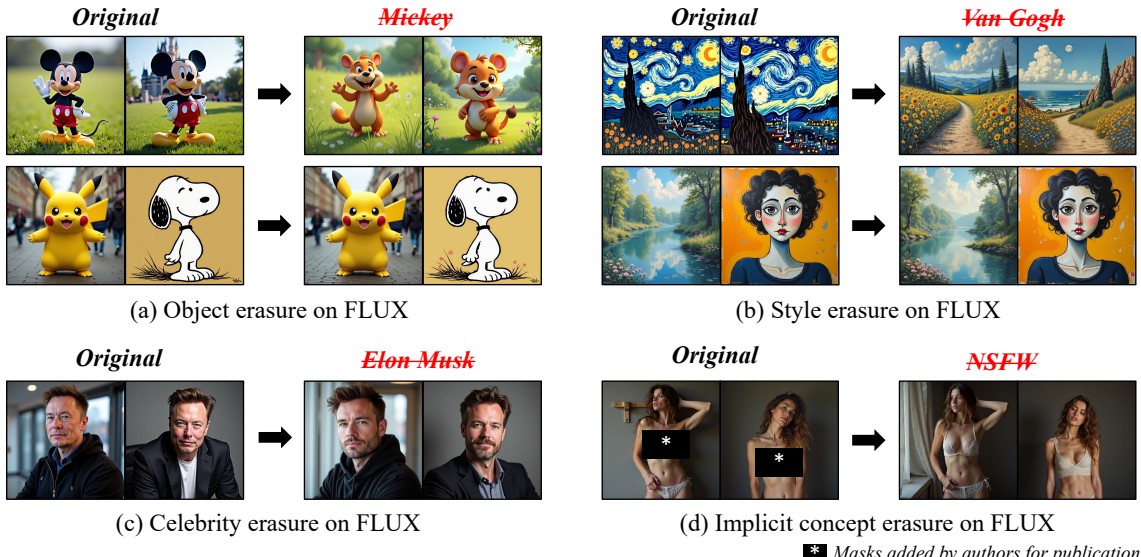

(a) Object erasure on FLUX      (b) Style erasure on FLUX

(c) Celebrity erasure on FLUX      (d) Implicit concept erasure on FLUX

⬛* *Masks added by authors for publication*

*Figure 4.* Extension to DiT-based models. All experiments are conducted on FLUX.1 dev. (a) Object erasure results for the target concept "*Mickey*", together with generations of non-target concepts ("*Pikachu*" and "*Snoopy*"). (b) Artistic style erasure results for "*Van Gogh*", together with non-target styles "*Monet*" and "*Picasso*". (c) Celebrity erasure results for "*Elon Musk*". (d) Implicit concept erasure results for NSFW content. Across all settings, our method achieves effective erasure while preserving non-target concepts, demonstrating strong transferability to DiT-based diffusion models.

not include the additional three preprocessing required for its preservation sets. By contrast, our method requires no such preprocessing and can be applied in a single-step manner, making it substantially more practical for large-scale multi-concept erasure. We also present qualitative results in Fig. 3, which show that our method maintains stable and effective erasure performance as the number of erased concepts increases, with minimal impact on non-target concepts compared to existing methods.

## 6.3. Implicit Concept Erasure

**Evaluation setup.** We evaluate our method on implicit concept erasure in scenarios where the target concept (e.g., nudity) is not explicitly appear in the prompt. Follow the settings in SPEED (Li et al., 2025c), we generate images with the Inappropriate Image Prompt (I2P) dataset (Schramowski et al., 2023) for evaluation, which consists of a diverse set of implicitly inappropriate prompts covering violence and sexual content. To further examine adversarial robustness, we evaluate all methods on two popular adversarial attack benchmarks, including Ring-A-Bell (Tsai et al., 2023) and MMA-Diffusion (Yang et al., 2024) . We use Nudenet with a threshold of 0.6 to detect sexual content and report the Attack Success Rate (ASR). We further generate images with the first 1K prompts from MSCOCO-30k, and report CS and FID to measure general preservation.

**Analysis and discussion.** We introduce additional baseline methods, including CPE (Lee et al., 2025), AdvUnlearn

(Zhang et al., 2024b), RACE (Kim et al., 2024), Receler (Huang et al., 2024), STEREO (Srivatsan et al., 2025) and EraseFlow (Kusumba et al., 2026), which defend against adversarial attacks with adversarial training objectives. Following SPEED (Li et al., 2025c), we also adapt our method with adversarial editing (denoted as Ours w/ AT) following the setting in RECE (Gong et al., 2024) for a fair comparison. As shown in Tab. 4, although some training-based methods (e.g., CPE) achieve strong robustness in concept erasure, they substantially degrade the generation quality of non-target concepts on COCO and incur prohibitive computational costs. In contrast, editing-based methods are generally more efficient, but often fail to provide sufficiently reliable erasure under such adversarial settings. Compared to these methods, our method can be seamlessly integrated with adversarial editing strategies, enabling robust concept erasure while effectively preserving the generation quality of unrelated concepts.

## 6.4. Further Analysis

**Extension to DiT-based Models.** To evaluate the transferability of our method, we further conduct experiments on the DiT-based model [2]FLUX.1 dev, covering object erasure, artistic style erasure, celebrity erasure, and implicit concept erasure. Unlike U-Net-based diffusion models, DiT-based architectures do not contain explicit cross-attention layers.

---

[2]https://github.com/black-forest-labs/flux

*Table 4.* Comparison under implicit concept erasure and adversarial attacks. Lower is better for I2P, MMA, Ring-A-Bell, and FID; higher is better for CS.

| Method | I2P ↓ | MMA ↓ | Ring-A-Bell ↓ | MSCOCO CS ↑ | FID ↓ |
|---|---|---|---|---|---|
| SD v1.4 | 0.52 | 0.63 | 0.98 | 26.32 | - |
| MACE | 0.21 | 0.04 | 0.05 | 24.06 | 52.78 |
| CPE | 0.07 | 0.01 | 0.00 | 26.32 | 48.23 |
| AdvUnlearn | 0.04 | 0.00 | 0.00 | 24.05 | 57.22 |
| UCE | 0.24 | 0.38 | 0.39 | 26.24 | 38.60 |
| RECE | 0.14 | 0.20 | 0.18 | 25.98 | 40.37 |
| RACE | 0.23 | 0.29 | 0.21 | 25.54 | 42.73 |
| Receler | 0.13 | 0.07 | 0.01 | 25.93 | 40.29 |
| SPEED | 0.20 | 0.24 | 0.20 | 26.29 | 37.82 |
| SPEED w/ AT | 0.10 | 0.01 | 0.00 | 26.03 | 39.51 |
| STEREO | 0.01 | 0.03 | 0.12 | 25.33 | 47.10 |
| EraseFlow | 0.04 | 0.02 | 0.10 | 25.11 | 42.33 |
| **Ours w/o AT** | 0.11 | 0.12 | 0.05 | 26.33 | 38.31 |
| **Ours w/ AT** | 0.05 | 0.01 | 0.00 | 26.10 | 39.73 |

*Table 5.* Ablation study on different objective formulations under the multi-concept erasure setting. **E** and **P** denote the erasure and preservation objective, respectively.

| Ablation | Subspace E | P | Erase 100 Celebrities $Acc_e$ ↓ | $Acc_s$ ↑ | $H_o$ ↑ | MSCOCO CS ↑ | FID ↓ |
|---|---|---|---|---|---|---|---|
| 1 | × | × | 7.59 | 91.37 | 91.70 | 25.83 | 20.79 |
| 2 | ✓ | ✓ | 4.54 | 93.01 | 94.22 | 26.17 | 18.10 |
| **Ours** | ✓ | × | 3.44 | 94.42 | **95.48** | 26.33 | 18.33 |
| SD v1.4 | – | – | 99.17 | 97.51 | 1.65 | 26.56 | – |

*Table 6.* Ablation study on the effect of the global preservation prior under the multi-concept erasure setting.

| $|C_g|$ ratio | Erase 100 Celebrities $Acc_e$ ↓ | $Acc_s$ ↑ | $H_o$ ↑ | MSCOCO CS ↑ | FID ↓ |
|---|---|---|---|---|---|
| None | 6.72 | 94.32 | 93.80 | 26.14 | 22.76 |
| 1/3 | 4.47 | 93.44 | 94.47 | 26.31 | 19.31 |
| 2/3 | 3.85 | 93.63 | 94.87 | 26.35 | 18.60 |
| **Full** | 3.44 | 94.42 | **95.48** | 26.33 | 18.33 |
| SD v1.4 | 99.17 | 97.51 | 1.65 | 26.56 | – |

Therefore, following [3]UCE, we choose to edit specific embedding layers in MMDiT.

As shown in Fig. 4, our method consistently achieves strong erasure performance on FLUX while effectively preserving non-target concepts, demonstrating its effectiveness across different diffusion architectures. Additional quantitative results are provided in the Appendix. D.2.

**Ablation on Objective Designs.** We conduct an ablation study to analyze the effect of different objective formulations under the multi-concept erasure setting. Specifically, we compare three variants: (1) vector-wise erasure with vector-wise preservation, (2) subspace-level erasure with subspace-level preservation, and (3) subspace-level erasure with vector-wise preservation (OCE). The results in Tab. 5 show that our asymmetric design achieves the best overall erasure performance. Specifically, subspace-level erasure mitigates interference among multiple erased concepts, while vector-wise preservation provides finer-grained control for preservation of non-target concepts. These results further validate the effectiveness of our asymmetric design.

**Ablation on global preservation prior $K_0$.** We study the effect of the size of the generic concept set used to construct $K_0$. Specifically, we evaluate reduced settings using 1/3 and 2/3 of the original COCO-30K tokens, as well as a variant without $K_0$ in the multi-concept erasure setting. In the latter case, we replace the original preservation term $\|PWC_0 - WC_0\|_F^2$ with $\|PWC_n - WC_n\|_F^2 + \|PW - W\|_F^2$, which changes the closed-form solution's preservation component to $W(I + C_nC_n^\top)W^\top$. As shown in Tab. 6, increasing the number of generic concepts consistently improves the

[3]https://github.com/rohitgandikota/unified-concept-editing

trade-off between erasure and preservation performance, demonstrating the effectiveness of the global preservation prior. The computation of $K_0$ is efficient ( 3s on A100 GPU) and performed offline.

## 7. Limitations.

Although OCE achieves effective erasure, it still has several limitations. First, the SVD-based subspace erasure may introduce additional computational overhead when scaling to larger models. Second, since OCE enforces subspace-level rather than strict vector-wise constraints, the generated outputs may fall into intermediate semantic regions instead of precisely matching fine-grained anchor concepts, which can limit its applicability in editing tasks. Finally, further exploration is needed for erasure of more implicit concepts such as relational, compositional understanding, or watermarks.

## 8. Conclusion

In this work, we propose **Orthogonal Concept Erasure (OCE)**, a geometry-driven editing-based method for concept erasure in text-to-image diffusion models. By revisiting concept erasure from a geometric perspective, OCE formulates concept erasure as a multiplicative orthogonal transformation applied to the parameters. This design enables direct control over neuron directions while preserving their intrinsic angular geometry. Moreover, by extending vector-wise erasure to a structured subspace-level projection objective with a closed-form solution, OCE achieves precise and stable erasure of target concepts while effectively retaining non-target concepts. This formulation naturally scales to multi-concept erasure and transfers to DiT-based models.

# Acknowledgments

This work is supported by the National Nature Science Foundation of China (62425114, 62121002, U23B2028, 62472395).

# Impact Statement

Our work proposes Orthogonal Concept Erasure (OCE), a method for efficiently and precisely removing sensitive or undesired concepts from text-to-image diffusion models while preserving the overall generative capabilities. OCE supports both single- and multi-concept erasure with minimal interference on unrelated concepts, providing a controllable mechanism to improve the safety and reliability of generative AI models. By reducing the risk of generating harmful, offensive, or privacy-violating content, OCE promotes safer AI deployment in content creation, education, and public platforms. This approach offers a practical and scalable solution for enhancing model controllability and aligning generative outputs with ethical and social standards.

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

# A. Additional Theoretical Analysis

## A.1. Derivation of the Vector-wise Orthogonal Erasure

We provide a detailed derivation of the closed-form solution to the vector-wise orthogonal erasure objective in Eq. (9).

Starting from the objective

$$\min_{P^\top P=I} \|PA - B\|_F^2, \tag{21}$$

we expand the Frobenius norm as

$$\|PA - B\|_F^2 = \mathrm{tr}((PA - B)^\top (PA - B)) \tag{22}$$
$$= \mathrm{tr}(A^\top A) + \mathrm{tr}(B^\top B) - 2\,\mathrm{tr}(P^\top BA^\top), \tag{23}$$

where we use the orthogonality constraint $P^\top P = I$.

Since $\mathrm{tr}(A^\top A)$ and $\mathrm{tr}(B^\top B)$ are constant with respect to $P$, minimizing the original objective is equivalent to maximizing

$$\max_{P^\top P=I} \mathrm{tr}(P^\top M), \qquad M = BA^\top. \tag{24}$$

This is the classical orthogonal Procrustes problem, whose optimal solution is given by

$$P = UV^\top, \tag{25}$$

where $U\Sigma V^\top$ is the singular value decomposition of $M$.

## A.2. Derivation of Subspace-level Orthogonal Concept Erasure (OCE)

In this appendix, we provide detailed derivations for the proposed subspace-level Orthogonal Concept Erasure (OCE) objective. In particular, we clarify the connection between vector-wise and subspace-level formulations, and rigorously justify the unified quadratic objective used to obtain the closed-form orthogonal update.

### A.2.1. SUBSPACE-LEVEL ERASURE OBJECTIVE

For erasure, instead of enforcing point-wise alignment between target and anchor embeddings, we operate at the subspace level. Let

$$G = \mathrm{orth}(WC_1), \qquad G_* = \mathrm{orth}(WC_*) \tag{26}$$

denote orthonormal bases for the target and anchor subspaces, respectively. Their corresponding projection matrices are

$$R = GG^\top, \qquad R_* = G_* G_*^\top. \tag{27}$$

We aim to suppress components of the transformed target subspace that lie outside the anchor subspace. This leads to the following erasure objective:

$$\min_{P^\top P=I} -\|PR - R_{*,\perp}\|_F^2. \tag{28}$$

Where $R_{*,\perp} = I - R_{*,\perp}$, Expanding the Frobenius norm and discarding constants gives

$$\|PR - R_{*,\perp}\|_F^2 = \mathrm{tr}(R^\top P^\top PR) - 2\,\mathrm{tr}(R_{*,\perp}^\top PR) + \mathrm{const} \tag{29}$$

Under the orthogonality constraint $P^\top P = I$, the first term is constant. Therefore, minimizing Eq. (28) is equivalent to the following linear trace maximization:

$$\max_{P^\top P=I} \mathrm{tr}(P^\top M_e), \qquad M_e = -(I - R_*)R. \tag{30}$$

A.2.2. VECTOR-WISE PRESERVATION OBJECTIVE

Recall that the vector-wise preservation objective for the retain set $C_0$ is defined as

$$\min_{P^\top P = I} \|PWC_0 - WC_0\|_F^2. \tag{31}$$

Expanding the squared Frobenius norm yields

$$\|PWC_0 - WC_0\|_F^2 = \text{tr}(C_0^\top W^\top P^\top PWC_0) - 2\,\text{tr}(C_0^\top W^\top PWC_0) + \text{const.} \tag{32}$$

Under the orthogonality constraint $P^\top P = I$, the first term is constant. Therefore, minimizing Eq. (31) is equivalent to the following linear trace maximization:

$$\max_{P^\top P = I} \text{tr}(P^\top M_0), \qquad M_0 = WC_0 C_0^\top W^\top, \tag{33}$$

where $M_0$ is symmetric and positive semidefinite.

Combining Eq. (33) and Eq. (30) leads to the final objective:

$$\max_{P^\top P = I} \text{tr}(P^\top (M_e + M_0)). \tag{34}$$

## B. Extented Related Works

In addition to the concept erasure methods discussed in the main text, there is also a large body of machine unlearning and concept erasure methods, including EA (Gao et al., 2025), MCE (Zhang et al., 2025), Co-Erasing (Li et al., 2025a), ANT (Li et al., 2025b), AGE (Bui et al., 2025), Realera (Liu et al., 2024), Salun (Fan et al., 2023), AdaVD (Wang et al., 2025), CPE (Lee et al., 2025), RACE (Kim et al., 2024), Receler (Huang et al., 2024), CURE (Biswas et al., 2025). Due to practical constraints on time and computational resources, it is infeasible to include exhaustive comparisons with all existing methods. Nevertheless, these methods provide valuable insights into efficient erasure, robustness, and controllability of diffusion models under different unlearning settings.

It is worth noting that the notion of *orthogonality* has also been explored in prior works such as AdaVD (Wang et al., 2025) and CURE (Biswas et al., 2025). However, these methods are fundamentally different from our approach. Specifically, AdaVD and CURE focus on *feature-level interventions during inference*, where orthogonal projections are employed as operational tools to suppress or filter target concepts at inference time. In contrast, our method represents a *paradigm-level innovation for editing-based concept erasure*. We study orthogonal updates from a *parameter-space geometric perspective*. We systematically analyze why orthogonal transformations are advantageous for concept erasure, and formulate a *rigorous erasure objective directly in the parameter space*. This formulation admits a *closed-form solution* and provides explicit theoretical guarantees, resulting in a more general, principled, and novel framework for concept erasure.

## C. Implementation Details

Our method introduces three hyperparameters $\lambda_e$, $\lambda_0$, and $\lambda_r$ to balance the contributions of erasure and preservation. The $M_{\text{total}}$ is formulated as

$$M_{\text{total}} = -\lambda_e R(I - R_*) + W(\lambda_0 K_0 + \lambda_r C_n C_n^\top) W^\top, \tag{35}$$

where the first term enforces suppression of the target concept, and the second term preserves generic ($K_0$) and neighboring ($C_n$) concepts, weighted respectively by $\lambda_0$ and $\lambda_r$. For all tasks, we edit the $W_k$ in the cross-attention modules for erasure.

*Table 7.* Object classes and corresponding anchors used in object erasure.

| **Object Classes** | Airplane | Automobile | Bird | Cat | Deer | Dog | Frog | Horse | Ship | Truck |
|---|---|---|---|---|---|---|---|---|---|---|
| **Anchors** | Sky | Truck | Cat | Dog | Horse | Cat | Bird | Deer | Airplane | Ship |

*Table 8.* Evaluation of Object Erasure on CIFAR-10. We report $Acc_e$ (lower indicates more successful erasure), $Acc_s$ (higher indicates better preservation of non-target concepts) and $H_o$ (higher indicates better overall performance) across the first five categories. All values are expressed as percentages (%). The original model's performance is included for reference.

| Concept | Dog | | | Frog | | | Horse | | | Ship | | | Truck | | | Average | | |
|---|---|---|---|---|---|---|---|---|---|---|---|---|---|---|---|---|---|---|
| | $Acc_e \downarrow$ | $Acc_s \uparrow$ | $H_o \uparrow$ | $Acc_e \downarrow$ | $Acc_s \uparrow$ | $H_o \uparrow$ | $Acc_e \downarrow$ | $Acc_s \uparrow$ | $H_o \uparrow$ | $Acc_e \downarrow$ | $Acc_s \uparrow$ | $H_o \uparrow$ | $Acc_e \downarrow$ | $Acc_s \uparrow$ | $H_o \uparrow$ | $Acc_e \downarrow$ | $Acc_s \uparrow$ | $H_o \uparrow$ |
| SD v1.4 | 98.74 | 98.62 | 2.49 | 99.93 | 98.49 | 0.14 | 99.78 | 98.50 | 0.44 | 98.64 | 98.63 | 2.68 | 98.89 | 98.60 | 2.20 | 99.20 | 98.57 | 1.59 |
| CA | 98.50 | 98.57 | 2.96 | 99.92 | 98.62 | 0.16 | 99.74 | 98.63 | 0.52 | 98.18 | 98.50 | 3.57 | 98.50 | 98.61 | 2.96 | 98.97 | 98.59 | 2.04 |
| ESD | 27.03 | 89.75 | 80.49 | 12.32 | 88.05 | 87.86 | 17.69 | 82.23 | 82.27 | 18.38 | 94.32 | 87.51 | 26.11 | 85.35 | 79.21 | 20.31 | 87.94 | 83.61 |
| FMN | 97.64 | 98.12 | 4.61 | 91.60 | 94.59 | 15.43 | 99.63 | 93.14 | 0.74 | 97.97 | 98.21 | 3.98 | 97.64 | 97.86 | 4.61 | 96.90 | 96.38 | 6.01 |
| UCE | 13.22 | 98.69 | 92.35 | 20.86 | 98.32 | 87.69 | 4.66 | 98.32 | 96.91 | 6.13 | 98.41 | 96.09 | 20.58 | 98.16 | 87.80 | 13.09 | 98.38 | 89.24 |
| MACE | 11.07 | 96.77 | 92.68 | 11.45 | 97.75 | 92.92 | 4.89 | 97.48 | 96.28 | 8.58 | 98.56 | 94.86 | 7.29 | 98.38 | 95.46 | 8.66 | 97.79 | 94.46 |
| RECE | 17.72 | 98.51 | 89.67 | 11.69 | 98.43 | 93.10 | 9.98 | 98.10 | 93.89 | 21.92 | 98.41 | 87.07 | 30.42 | 98.43 | 81.53 | 18.35 | 98.38 | 89.24 |
| SPEED | 1.32 | 98.43 | 98.55 | 5.63 | 98.18 | 96.24 | 3.84 | 97.83 | 96.99 | 0.78 | 98.44 | 98.83 | 7.49 | 98.55 | 95.43 | 3.81 | 98.29 | 97.23 |
| **Ours** | 0.74 | 98.47 | **98.86** | 1.27 | 98.45 | **98.59** | 0.35 | 98.39 | **99.02** | 0.15 | 98.64 | **99.24** | 1.38 | 98.53 | **98.57** | 0.78 | 98.50 | **98.86** |

*Table 9.* Quantitative experiments on implicit concept erasure with FLUX.1 dev. Lower is better for I2P, Ring-A-Bell, and FID; higher is better for CS.

| Method | I2P $\downarrow$ | Ring-A-Bell $\downarrow$ | MSCOCO | |
|---|---|---|---|---|
| | | | CS $\uparrow$ | FID $\downarrow$ |
| FLUX | 0.44 | 0.93 | 25.64 | – |
| UCE | 0.30 | 0.56 | 25.63 | 30.41 |
| ESD | 0.23 | 0.61 | 24.97 | 55.43 |
| EraseFlow | 0.21 | 0.44 | 24.72 | 40.64 |
| **Ours** | 0.23 | 0.43 | 25.55 | 26.29 |

## C.1. Experimental Setup Details

**Single-concept erasure.** For object erasure, we conduct experiments on the ten objects classes of the CIFAR-10 dataset, as detailed in Tab. 7. For erasure of each class, we edit the original model with $\lambda_e = 1000$, $\lambda_0 = 50$, and $\lambda_r = 1$. In evaluation, 200 images are generated using the prompt "a photo of the *class*" for the target concept and the nine remaining concepts. For artistic style erasure, we conduct experiments on three style: Van Gogh, Monet and Picasso. we edit the original model with $\lambda_e = 500$, $\lambda_0 = 100$, and $\lambda_r = 1$. In evaluation, 100 images are generated with the prompt "a painting in the style of *style*" for the target style.

**Multi-concept erasure.** Following the experiment setup of SPEED (Li et al., 2025c), we introduce a dataset of 200 celebrities, whose portraits generated by SD v1.4 can be reliably recognized with high accuracy by the GIPHY Celebrity Detector (GCD) (Hasty et al., 2019). We divide this dataset into an erasure set containing 100 celebrities, and a retain set of 100 other celebrities. We provide the full lists for both sets in Tab. 13. We conduct experiments erasing 10, 50, and 100 celebrities from the predefined erasure set, while including the full retain set. For erasing 10 celebrities, we set the hyperparameters as $\lambda_e = 1500$, $\lambda_0 = 50$, and $\lambda_r = 3$. For erasing 50 celebrities, we set $\lambda_e = 1200$, $\lambda_0 = 50$, and $\lambda_r = 3$ For the 100-celebrity erasure scenario, we set $\lambda_e = 900$, $\lambda_0 = 50$, and $\lambda_r = 3$. In evaluation, we use five celebrity templates: "a portrait of *celebrity*", "a sketch of *celebrity*", "an oil painting of *celebrity*", "*celebrity* in an official photo", and "an image capturing **celebrity** at a public event". We generate 500 images for each set. For non-target concepts, we generate 1 image per template for each of the 100 concepts. For target concepts, the number of images per template is adjusted to maintain 500 images overall.

**Implementation of baselines.** We compare OCE with seven baselines: [4]CA (Kumari et al., 2023), [5]ESD (Gandikota et al., 2023), [6]FMN (Zhang et al., 2024a), [7]UCE (Gandikota et al., 2024), [8]MACE (Lu et al., 2024), [9]RECE(Gong et al.,

---

[4]https://github.com/nupurkmr9/concept-ablation
[5]https://github.com/rohitgandikota/erasing
[6]https://github.com/SHI-Labs/Forget-Me-Not
[7]https://github.com/rohitgandikota/unified-concept-editing
[8]https://github.com/Shilin-LU/MACE
[9]https://github.com/CharlesGong12/RECE

*Table 10.* Ablation study on selection of anchor concepts on CIFAR-10.

| Concept | Airplane | | | Automobile | | | Bird | | | Cat | | | Deer | | | Average | | |
|---|---|---|---|---|---|---|---|---|---|---|---|---|---|---|---|---|---|---|
| | $Acc_e \downarrow$ | $Acc_s \uparrow$ | $H_o \uparrow$ | $Acc_e \downarrow$ | $Acc_s \uparrow$ | $H_o \uparrow$ | $Acc_e \downarrow$ | $Acc_s \uparrow$ | $H_o \uparrow$ | $Acc_e \downarrow$ | $Acc_s \uparrow$ | $H_o \uparrow$ | $Acc_e \downarrow$ | $Acc_s \uparrow$ | $H_o \uparrow$ | $Acc_e \downarrow$ | $Acc_s \uparrow$ | $H_o \uparrow$ |
| Empty | 9.22 | 97.63 | 94.08 | 12.61 | 97.13 | 92.00 | 3.12 | 97.90 | 97.38 | 5.03 | 98.41 | 96.66 | 6.90 | 98.47 | 95.71 | 7.38 | 97.91 | 95.17 |
| Random | 7.71 | 98.62 | 95.35 | 10.91 | 98.56 | 93.59 | 5.62 | 97.36 | 95.85 | 2.49 | 98.55 | 98.02 | 4.28 | 98.32 | 97.00 | 6.20 | 98.28 | 95.96 |
| **Ours** | 6.89 | 98.85 | **95.89** | 8.79 | 99.07 | **94.98** | 3.40 | 98.30 | **97.44** | 0.15 | 98.71 | **99.28** | 3.81 | 98.45 | **97.31** | 4.61 | 98.68 | **96.98** |

*Table 11.* Ablation study on selection of anchor concepts in multi-concept erasure scenarios.

| Anchor | Erase 100 Celebrities | | | MSCOCO | |
|---|---|---|---|---|---|
| | $Acc_e \downarrow$ | $Acc_s \uparrow$ | $H_o \uparrow$ | CS$\uparrow$ | FID$\downarrow$ |
| Empty | 4.28 | 91.58 | 93.60 | 26.04 | 18.64 |
| Car | 4.48 | 94.68 | 95.10 | 25.85 | 18.18 |
| Cat | 4.24 | 92.20 | 93.95 | 25.92 | 18.45 |
| Person | 3.85 | 94.50 | 95.32 | 26.40 | 18.29 |
| **Ours** | 3.44 | 94.42 | **95.48** | 26.33 | 18.33 |

2024), [10]SPEED (Li et al., 2025c). All the baselines are implemented using the default configurations from their official repositories.

## D. Additional Experiments

### D.1. Additional Evaluation Results of Object Erasure on CIFAR-10

We further present the results of erasing the remaining five object classes of the CIFAR-10 dataset in Tab. 8. OCE achieves the highest $H_o$ in all erasure of these five class of objects. This demonstrates that OCE consistently maintains a superior balance between effective concept erasure and preservation of non-target generation capacity.

### D.2. Additional Experiments on FLUX

We conducted additional quantitative experiments on implicit concept erasure with FLUX.1 dev. We generate images with the I2P and Ring-A-Bell dataset for evaluation. The results are summarized in Tab. 9. Compared with baseline methods, OCE achieves a better balance between concept erasure and prior preservation, demonstrating its strong transferability to different model architectures. For all baseline methods, we use the official implementations with default settings for fair comparison.

### D.3. Ablation on Selection of Anchor Concepts

For anchor selection, we adopt a heuristic strategy: anchor concepts are chosen to belong to the same high-level category as the target concept, sharing some similarity but also exhibiting noticeable differences. This ensures smooth and stable concept erasure. To further investigate the impact of anchor choice, we conducted experiments on CIFAR-10 (in Tab. 10) and multi-concept erasure scenarios (in Tab. 11). We compared our heuristic selections with random anchors and the empty anchor. The performance varies under different anchor settings, and our heuristic strategy consistently improves erasure effectiveness.

### D.4. Ablation on the Hyperparameters

We conduct an ablation study on the trade-off parameters $\lambda_e$, $\lambda_0$, and $\lambda_r$ in the multi-concept erasure scenario. The results are summarized in the Tab. 12. From the table, we can observe:$\lambda_e$ mainly affects the erasure effectiveness ($Acc_e$),$\lambda_0$ mainly affects the overall generation quality (FID(COCO)),and $\lambda_p$ mainly affects the preservation of neighboring concepts ($Acc_r$).These findings are consistent with our design. In our experiments, we choose $\lambda_e = 900$, $\lambda_0 = 50$, and $\lambda_r = 3$ to achieve a balanced trade-off between erasure and preservation.

[10]https://github.com/ouxiang-li/speed

*Table 12.* Ablation study on hyperparameters in multi-concept erasure scenarios.

| $\lambda_e$ | $\lambda_0$ | $\lambda_r$ | Erase 100 Celebrities | | | MSCOCO | |
| | | | $\text{Acc}_e \downarrow$ | $\text{Acc}_s \uparrow$ | $H_o \uparrow$ | CS $\uparrow$ | FID $\downarrow$ |
|---|---|---|---|---|---|---|---|
| **900** | **50** | **3** | 3.44 | 94.42 | **95.48** | 26.33 | 18.33 |
| 1000 | 50 | 3 | 2.85 | 93.25 | 95.16 | 26.29 | 19.01 |
| 1200 | 50 | 3 | 2.03 | 89.73 | 93.67 | 26.24 | 19.69 |
| 600 | 50 | 3 | 7.70 | 93.48 | 92.89 | 26.41 | 17.61 |
| 900 | 20 | 3 | 3.05 | 94.01 | 95.46 | 26.12 | 21.05 |
| 900 | 80 | 3 | 5.67 | 93.85 | 94.09 | 26.44 | 17.55 |
| 900 | 50 | 1 | 1.84 | 89.57 | 93.67 | 26.34 | 18.63 |
| 900 | 50 | 5 | 6.52 | 94.49 | 93.98 | 26.37 | 18.34 |

## D.5. Additional Qualitative Results

In this section, we provide more comprehensive qualitative results across a variety of concept erasure tasks, including object erasure (Fig. 5), style erasure (Fig. 6), celebrity erasure (Fig. 7) and implicit concept erasure (Fig. 8).

*Table 13.* Evaluation setup for multi-concept erasure. The dataset contains an Erasure Set and a Retain Set of celebrities.

| Group | Number | Anchor Concept | Celebrity |
|---|---|---|---|
| Erasure Set | 10 | 'celebrity' | *'Adam Driver', 'Adriana Lima', 'Amber Heard', 'Amy Adams', 'Andrew Garfield', 'Angelina Jolie', 'Anjelica Huston', 'Anna Faris', 'Anna Kendrick', 'Anne Hathaway'* |
| | 50 | 'celebrity' | *'Adam Driver', 'Adriana Lima', 'Amber Heard', 'Amy Adams', 'Andrew Garfield', 'Angelina Jolie', 'Anjelica Huston', 'Anna Faris', 'Anna Kendrick', 'Anne Hathaway', 'Arnold Schwarzenegger', 'Barack Obama', 'Beth Behrs', 'Bill Clinton', 'Bob Dylan', 'Bob Marley', 'Bradley Cooper', 'Bruce Willis', 'Bryan Cranston', 'Cameron Diaz', 'Channing Tatum', 'Charlie Sheen', 'Charlize Theron', 'Chris Evans', 'Chris Hemsworth', 'Chris Pine', 'Chuck Norris', 'Courteney Cox', 'Demi Lovato', 'Drake', 'Drew Barrymore', 'Dwayne Johnson', 'Ed Sheeran', 'Elon Musk', 'Elvis Presley', 'Emma Stone', 'Frida Kahlo', 'George Clooney', 'Glenn Close', 'Gwyneth Paltrow', 'Harrison Ford', 'Hillary Clinton', 'Hugh Jackman', 'Idris Elba', 'Jake Gyllenhaal', 'James Franco', 'Jared Leto', 'Jason Momoa', 'Jennifer Aniston', 'Jennifer Lawrence'* |
| | 100 | 'celebrity' | *'Adam Driver', 'Adriana Lima', 'Amber Heard', 'Amy Adams', 'Andrew Garfield', 'Angelina Jolie', 'Anjelica Huston', 'Anna Faris', 'Anna Kendrick', 'Anne Hathaway', 'Arnold Schwarzenegger', 'Barack Obama', 'Beth Behrs', 'Bill Clinton', 'Bob Dylan', 'Bob Marley', 'Bradley Cooper', 'Bruce Willis', 'Bryan Cranston', 'Cameron Diaz', 'Channing Tatum', 'Charlie Sheen', 'Charlize Theron', 'Chris Evans', 'Chris Hemsworth', 'Chris Pine', 'Chuck Norris', 'Courteney Cox', 'Demi Lovato', 'Drake', 'Drew Barrymore', 'Dwayne Johnson', 'Ed Sheeran', 'Elon Musk', 'Elvis Presley', 'Emma Stone', 'Frida Kahlo', 'George Clooney', 'Glenn Close', 'Gwyneth Paltrow', 'Harrison Ford', 'Hillary Clinton', 'Hugh Jackman', 'Idris Elba', 'Jake Gyllenhaal', 'James Franco', 'Jared Leto', 'Jason Momoa', 'Jennifer Aniston', 'Jennifer Lawrence', 'Jennifer Lopez', 'Jeremy Renner', 'Jessica Biel', 'Jessica Chastain', 'John Oliver', 'John Wayne', 'Johnny Depp', 'Julianne Hough', 'Justin Timberlake', 'Kate Bosworth', 'Kate Winslet', 'Leonardo Dicaprio', 'Margot Robbie', 'Mariah Carey', 'Melania Trump', 'Meryl Streep', 'Mick Jagger', 'Mila Kunis', 'Milla Jovovich', 'Morgan Freeman', 'Nick Jonas', 'Nicolas Cage', 'Nicole Kidman', 'Octavia Spencer', 'Olivia Wilde', 'Oprah Winfrey', 'Paul Mccartney', 'Paul Walker', 'Peter Dinklage', 'Philip Seymour Hoffman', 'Reese Witherspoon', 'Richard Gere', 'Ricky Gervais', 'Rihanna', 'Robin Williams', 'Ronald Reagan', 'Ryan Gosling', 'Ryan Reynolds', 'Shia Labeouf', 'Shirley Temple', 'Spike Lee', 'Stan Lee', 'Theresa May', 'Tom Cruise', 'Tom Hanks', 'Tom Hardy', 'Tom Hiddleston', 'Whoopi Goldberg', 'Zac Efron', 'Zayn Malik'* |
| Retain Set | 10, 50, and 100 | - | *'Aaron Paul', 'Alec Baldwin', 'Amanda Seyfried', 'Amy Poehler', 'Amy Schumer', 'Amy Winehouse', 'Andy Samberg', 'Aretha Franklin', 'Avril Lavigne', 'Aziz Ansari', 'Barry Manilow', 'Ben Affleck', 'Ben Stiller', 'Benicio Del Toro', 'Bette Midler', 'Betty White', 'Bill Murray', 'Bill Nye', 'Britney Spears', 'Brittany Snow', 'Bruce Lee', 'Burt Reynolds', 'Charles Manson', 'Christie Brinkley', 'Christina Hendricks', 'Clint Eastwood', 'Countess Vaughn', 'Dakota Johnson', 'Dane Dehaan', 'David Bowie', 'David Tennant', 'Denise Richards', 'Doris Day', 'Dr Dre', 'Elizabeth Taylor', 'Emma Roberts', 'Fred Rogers', 'Gal Gadot', 'George Bush', 'George Takei', 'Gillian Anderson', 'Gordon Ramsey', 'Halle Berry', 'Harry Dean Stanton', 'Harry Styles', 'Hayley Atwell', 'Heath Ledger', 'Henry Cavill', 'Jackie Chan', 'Jada Pinkett Smith', 'James Garner', 'Jason Statham', 'Jeff Bridges', 'Jennifer Connelly', 'Jensen Ackles', 'Jim Morrison', 'Jimmy Carter', 'Joan Rivers', 'John Lennon', 'Johnny Cash', 'Jon Hamm', 'Judy Garland', 'Julianne Moore', 'Justin Bieber', 'Kaley Cuoco', 'Kate Upton', 'Keanu Reeves', 'Kim Jong Un', 'Kirsten Dunst', 'Kristen Stewart', 'Krysten Ritter', 'Lana Del Rey', 'Leslie Jones', 'Lily Collins', 'Lindsay Lohan', 'Liv Tyler', 'Lizzy Caplan', 'Maggie Gyllenhaal', 'Matt Damon', 'Matt Smith', 'Matthew Mcconaughey', 'Maya Angelou', 'Megan Fox', 'Mel Gibson', 'Melanie Griffith', 'Michael Cera', 'Michael Ealy', 'Natalie Portman', 'Neil Degrasse Tyson', 'Niall Horan', 'Patrick Stewart', 'Paul Rudd', 'Paul Wesley', 'Pierce Brosnan', 'Prince', 'Queen Elizabeth', 'Rachel Dratch', 'Rachel Mcadams', 'Reba Mcentire', 'Robert De Niro'* |

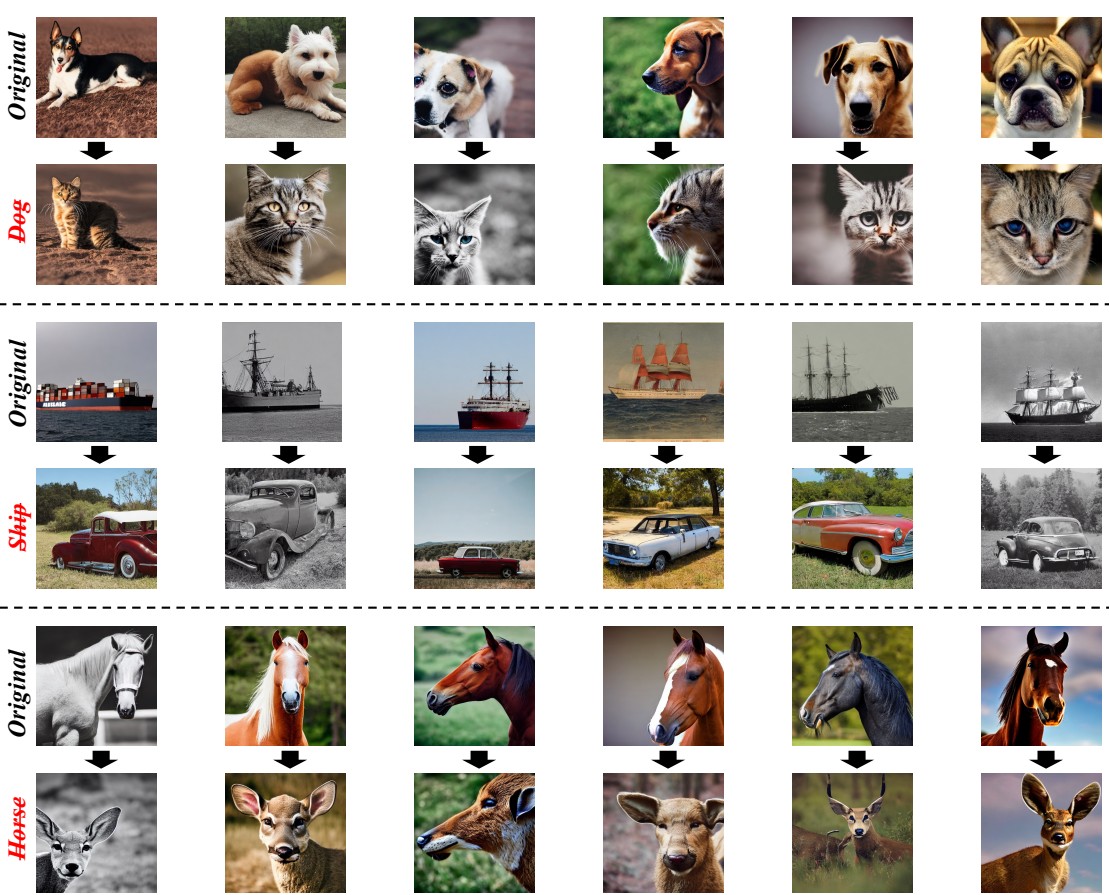

*Figure 5.* Additional qualitative results on Object Erasure.

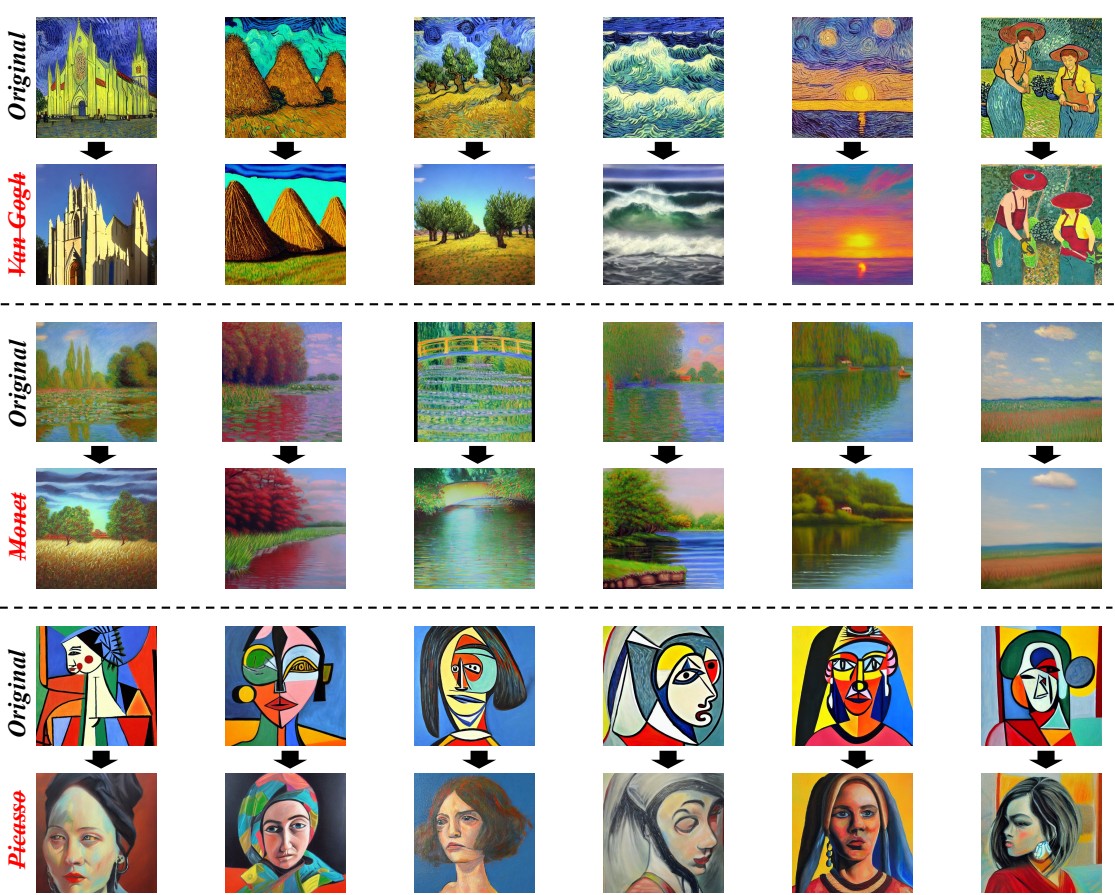

*Figure 6.* Additional qualitative results on Artistic Style Erasure.

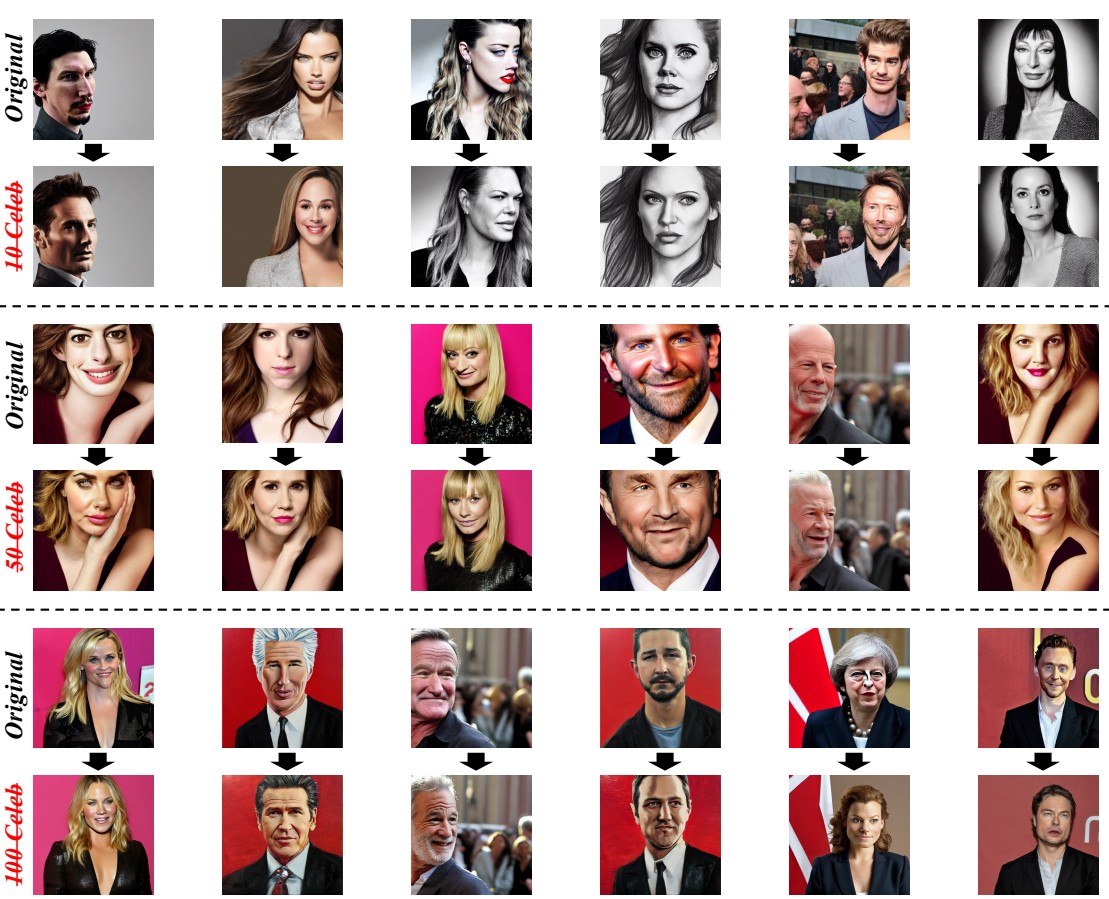

*Figure 7.* Additional qualitative results on Celebrity Erasure.

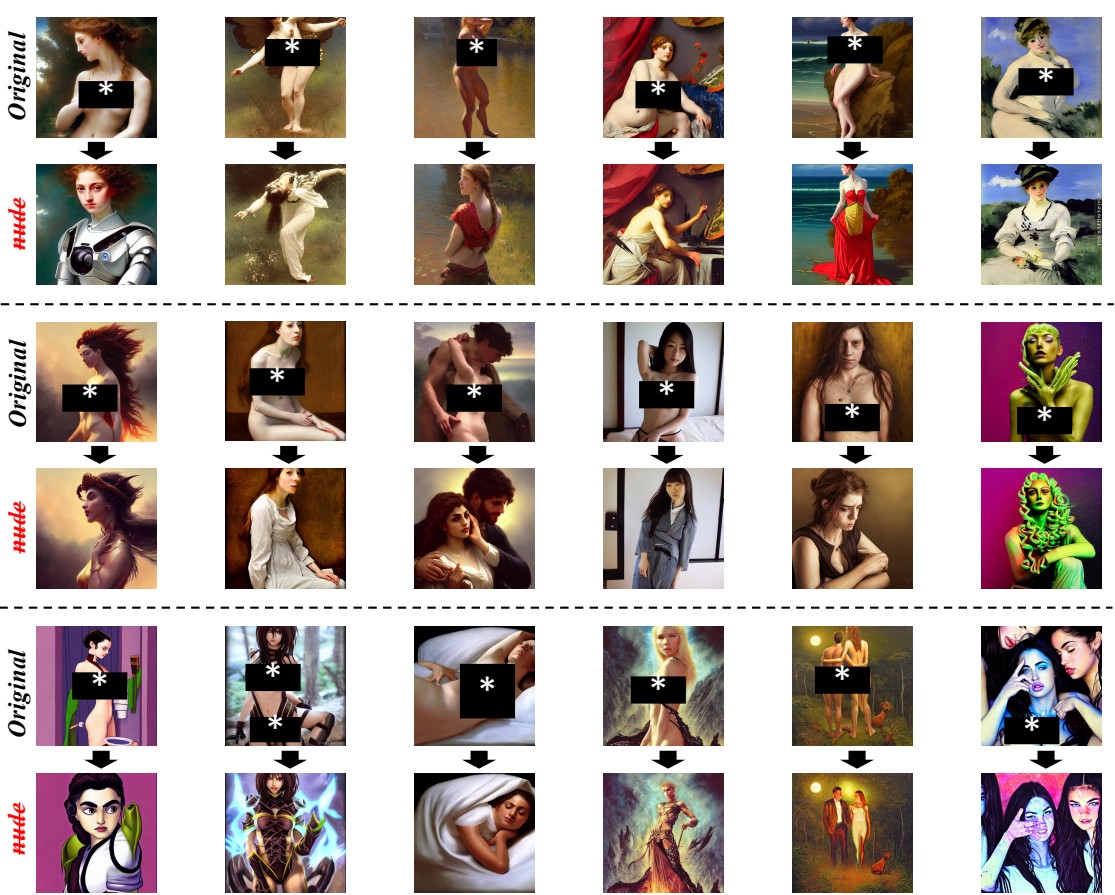

*Figure 8.* Additional qualitative results on Implicit Concept Erasure on I2P.

