# OpenReview forum: "Orthogonal Concept Erasure for Diffusion Models"
_ICML.cc/2026/Conference — ICML 2026 spotlight_

### Official Review · Reviewer_Lt6z · 2026-02-20

**Soundness:** 2
**Presentation:** 3
**Significance:** 3
**Originality:** 3
**Overall Recommendation:** 5
**Confidence:** 4

**Summary:**

This paper proposes a new closed-form concept erasure approach for diffusion models. In contrast to the "additive" approach of prior work like UCE, the newly proposed Orthogonal Concept Erasure (OCE) applies multiplicative parameter updates to better preserve important geometrical properties in the latent space of concept embedding projections. Their approach is motivated by a tiny toy experiment that demonstrates that the "neuron magnitude" carries the semantic information, while the "inter-neuron angles" are important for the utility of the model. The argument is that additive weight updates (like UCE) can affect the inter-neuron angular geometry and thereby degrade the model's utility more. Instead, OCE restricts its weight transformation to an orthogonal matrix, which preserves angles and lengths but only changes the semantically important angle of the space.

**Compliance With Llm Reviewing Policy:**

Affirmed.

**Final Justification:**

This paper presents an interesting and novel approach to concept erasure, which replaces additive updates in closed-form approaches like UCE with multiplicative updates restricted to orthogonal transformations, hence the name Orthogonal Concept Erasure (OCE). It's demonstrated to work on newer models like FLUX and better preserve utility than UCE. Since UCE inspired many later works with its closed-form approach, I think OCE could have a similar impact in the community when combined with other techniques, e.g., in a MACE-like framework to improve scalability. Overall, all of my key concerns were addressed in the rebuttal, and I increased my rating from 4 to **5 (Accept)**.

**Key Questions For Authors:**

- **Q1**: How was OCE applied to FLUX.1 [dev] even though FLUX does not have cross-attention layers? How was the OCE method adapted for the DiT-based architecture? Am I right to assume that the migration was done analogously to how UCE is applied to FLUX?
- **Q2**: How does OCE compare to UCE, ESD, and EraseFlow quantitatively and qualitatively on FLUX?
- **Q3**: In lines 81 to 85, it is argued that editing-based methods (in your context: UCE, RECE, and SPEED) struggle to reliably preserve prior concepts and rely on complex erasure pipelines. That's a passage I do not understand. These closed-form scalable approaches often allow explicit preservation concepts and are per design not complex and fast. I would like to understand your reasoning behind this argument.

**Limitations:**

I could not find any discussion of limitations or potential negative impacts. I recommend adding a few sentences on the limitations of closed-form approaches, such as architectural constraints, to round out this work.

**Strengths And Weaknesses:**

**Soundness & Presentation**:  This paper is nicely written with a good flow and a formal derivation (the one in the main paper) of the proposed method looks sound. Overall, this work opens a new interesting perspective on the problem. Unfortunately, I do have some concerns about the positioning and quantitative evaluation:
- All quantitative experiments in this work are on Stable Diffusion 1 (SD 1), which by now is largely outdated, while newer flow matching model families like FLUX or Qwen-Image are the state of the art. The paper does not resolve whether the intuition from the toy experiment also transfers to FLUX.
- OCE is a closed-form approach to concept erasure, which restricts it architecturally to certain layers in a model that it can be applied to, presumably just like UCE (Gandikota et al. WACV'24).
- Being aware that UCE can also be applied to FLUX by choosing only very specific layers *before* the DiT (as compared to CA layers within the U-Net of SD 1), the main experiments of this work should have been executed on the newer types of models. The results in Figure 4 of the main paper are only qualitative.
- Some more recent non-closed-form baselines should be considered: STEREO (Srivatsan et al. CVPR'25) and EraseFlow (Kusumba et al. NeurIPS'25 Spotlight)
- Generally, the robustness dimension is not considered in the main paper.
- Table 8 in the Appendix misses the row of the original model.

**Significance**: Suppressing certain content from text-to-image generation models is a timely area of research. Closed-form approaches are better at being efficient and scalable in comparison to more involved finetuning methods like ESD or STEREO, which tend to be not very scalable and more fatal for the model's utility when scaled up to multiple concepts. So, proposing a new, even better closed-form approach that beats all baselines in the challenging scenario of erasing 100 celebrities is significant, even though the concerns on the generalizability of these results to SOTA model families like FLUX remain.

Another point I want to raise is that the main focus is on explicit erasure of objects and celebrities; probably because these scenarios often work better for closed-form rewiring approaches like UCE or OCE. Arguably, the broader, more practical applications like the erasure of explicit content is only covered in the Appendix D.2 under "Implicit Concept Erasure". There, the robustness dimension is part of the evaluation suite with MMA, RAB, and UnlearnDiff (prompts that were adversarially designed to circumvent erasure). Interestingly, OCE is more robust than UCE (e.g., on RAB), which is a finding that I think is worth more experiments and then potentially a place in the main paper.

**Originality**: This work definitely offers a novel perspective on performing concept erasure in a closed-form manner.

---

> ### Author Rebuttal · Authors · 2026-03-31
>
> > **W1: Toy experiments on FLUX**
>
> Thank you for the suggestion. We conducted additional toy experiments on the FLUX.1-dev to verify the transferability of our insights. Following the same setup as in the toy SD experiments, we observed similar phenomena in this DiT-based architecture, confirming that the model geometry plays a key role in concept erasure tasks. The qualitative results are shown in https://anonymous.4open.science/r/OCE_rebuttal-7006
>
> >**Q1: How was OCE applied to FLUX.1 [dev]**
>
> Thank you for the question. To adapt OCE to FLUX.1-dev, we followed the publicly official UCE implementation on FLUX and updated the *context_embedder* and *text_embedder.linear_1* modules instead. We note that exploring more effective migration strategies fis an interesting direction for future work.
>
> > **W2 & Q2: Compare to UCE, ESD, and EraseFlow on FLUX**
>
> Thank you for the suggestion. We conducted additional quantitative experiments on NSFW concept erasure on Flux.1-dev, and the results are summarized in the table below. Compared with baseline methods, OCE achieves a better balance between concept erasure and prior preservation, demonstrating its strong transferability to different model architectures. For all baseline methods, we use the official implementations with default settings for fair comparison. The qualitative results are shown in https://anonymous.4open.science/r/OCE_rebuttal-7006
>
> | Method      | I2P      | Ring-A-Bell     | FID(COCO)   | CLIP(COCO) |
> |:-----------:|:--------:|:--------:|:-----:|:-----:|
> | FLUX        | 0.44     | 0.93     | -     | 25.64 |
> | UCE        | 0.30     | 0.56     | 30.41 | 25.63 |
> | ESD         | 0.23     | 0.61     | 55.43 | 24.97 |
> | EraseFlow   | 0.21     | 0.44     | 40.64 | 24.72 |
> | Ours         | 0.23     | 0.43     | 26.29 | 25.55 |
>
> >**W3: Compare to STEREO and EraseFlow on SD 1.4**
>
> Thank you for the suggestion. We conduct additional comparisons with STEREO and EraseFlow on two tasks: NSFW concept erasure and “Van Gogh” style erasure. Our method achieves competitive erasure performance compared to STEREO and EraseFlow while better preserving generation quality. Moreover, our closed-form method  is more computationally efficient than these optimization-based approaches. We will include these results and discussions in the revised version.
>
> *NSFW*
> |Method|I2P|MMA|Ring-A-Bell|UnlearnDiff|FID(COCO)|CS(COCO)|
> |:-:|:-:|:-:|:-:|:-:|:-:|:-:|
> |SD v1.4|0.52|0.63|0.98|0.98|-|26.32|
> |STEREO|0.01|0.03|0.12|0.32|47.10|25.33|
> |EraseFlow|0.04|0.02|0.10|0.35|42.33|25.11|
> |Ours w/o AT|0.11|0.12|0.05|0.65|38.31|26.33|
> |Ours w/ AT|0.05|0.01|0.00|0.40|39.73|26.10|
>
> *Van Gogh*
> |Method|CS|FID(COCO)|CS(COCO)|
> |:-:|:-:|:-:|:-:|
> |SD v1.4|25.51|-|26.56|
> |STEREO|19.58|14.48|25.89|
> |EraseFlow|22.70|14.22|26.42|
> |Ours|21.08|7.15|26.52|
>
> > **W4: Explicit contents erasure and robustness**
>
> Thank you for the suggestion. Due to page limitations, discussions on explicit content erasure were included in the appendix. We will move this part to the main paper and include the above analyses in the revised version. Regarding robustness, OCE consistently outperforms UCE. We attribute this to the fact that OCE enforces constraints at the subspace level, which provides more stable and global control over the embedding transformations, leading to improved robustness. In contrast, UCE applies vector-wise erasure with relatively aggressive additive updates, which may disrupt the geometric structure of the model’s parameter space and result in less stable erasure behavior.
>
> >**Q3: Clarification on prior-preservation and complexity of existing closed-form editing methods**
>
> Thank you for the question. While existing closed-form methods allow explicit preservation concepts, experiments show that they can still significantly affect the model’s generative ability. This aligns with our analysis that many prior erasure methods do not maintain the underlying model geometry, leading to suboptimal preservation.
>
> Regarding complex pipelines, SOTA methods like SPEED achieve better preservation compared to UCE or RECE, but at the cost of additional preprocessing steps such as Influence-based Prior Filtering, Directed Prior Augmentation, and Invariant Equality Constraints, which introduce extra overhead outside of training. In contrast, our method leverages model geometry to provide a stable and principled erasure paradigm, achieving efficient erasure and excellent prior preservation without any complex auxiliary procedures. We will revise the manuscript to improve the clarity of this discussion.
>
> > **Lack of Limitations**
>
> Thank you for the suggestion. We acknowledge the importance of discussing limitations and failure modes, and we will include them in the revised manuscript.

---

> > ### Author Rebuttal · Reviewer_Lt6z · 2026-04-01
> >
> > I want to thank the authors for their targeted rebuttal with additional experiments and helpful answers to my open questions. Overall, my key concerns were addressed, and I support the acceptance of this paper and will increase my score accordingly.
> >
> > Thanks for this work!

---

> > > ### Author Response · Authors · 2026-04-02
> > >
> > > Thank you for the encouraging feedback and support. We're happy that the additional experiments and clarifications helped address your concerns, and we truly appreciate your helpful suggestions.

---

### Official Review · Reviewer_kex6 · 2026-03-09

**Soundness:** 3
**Presentation:** 4
**Significance:** 3
**Originality:** 3
**Overall Recommendation:** 5
**Confidence:** 4

**Summary:**

This paper proposes Orthogonal Concept Erasure (OCE), a geometry-driven editing-based method for removing undesired concepts from text-to-image diffusion models. Based on empirical analysis from a geometric perspective of model editing, the authors formulate concept erasure as layer-wise orthogonal transformations with closed-form solutions. They further introduce a subspace-level objective for concept retention. Experiments demonstrate superior erasure effectiveness while preserving non-target concepts.

**Compliance With Llm Reviewing Policy:**

Affirmed.

**Final Justification:**

The method is technically sound, the contribution is significant for the concept erasure community, and the additional experiments demonstrate good generalizability to modern architectures. Therefore, I recommend acceptance.

**Key Questions For Authors:**

(Q1) Are there any constraints, issues, or concerns when the proposed method is applied to non-UNet models compared with other baselines? This question is related to Weakness 1.

(Q2) In Table 4, how is the universal retention set used to construct $C_g$ prepared? I am interested in the relationship between the number of concepts included in $C_g$ and the performance in terms of generation quality and specificity.

(Q3) I could not find any description of the limitations and failure modes of the proposed method. Could the authors clarify these aspects?

**Limitations:**

Although impact statement is described, the discussion of technical limitation is not sufficient.

**Strengths And Weaknesses:**

### Strength

- The authors provide effective empirical evidence to support their geometric hypothesis, which offers a novel insight into concept erasure.

- The subspace-level erasure approach is well-motivated (preservation requires finer-grained control) and mathematically elegant.

- Despite the advanced proposed components, the method still admits a closed-form solution, ensuring computational efficiency.

- The proposed method is compared against current baselines across multiple empirical setups, achieving impressive performance.

- The paper is very easy to follow, with a clear step-by-step presentation of the method.


### Weaknesses

(W1) Although the proposed method is applied to FLUX, there are no quantitative results for this experiment. Readers would be interested in seeing whether the method is competitive with or outperforms other baselines on such modern architectures.

(W2) While the ablation study (Table 4) demonstrates the effectiveness of the subspace technique, more in-depth analysis would be appreciated (please see the question section).

---

> ### Author Rebuttal · Authors · 2026-03-31
>
> >**W1 & Q1: More results and discussions on FLUX**
>
> Thank you for the suggestion. We conducted additional quantitative experiments on NSFW concept erasure on FLUX.1-dev, and the results are summarized in the table below. Compared with baseline methods, OCE achieves a better balance between concept erasure and prior preservation, demonstrating its strong transferability to different model architectures.
>
> | Method      | I2P      | Ring-A-Bell     | FID(COCO)   | CLIP(COCO) |
> |:-----------:|:--------:|:--------:|:-----:|:-----:|
> | FLUX        | 0.44     | 0.93     | -     | 25.64 |
> | UCE        | 0.30     | 0.56     | 30.41 | 25.63 |
> | ESD         | 0.23     | 0.61     | 55.43 | 24.97 |
> | EraseFlow   | 0.21     | 0.44     | 40.64 | 24.72 |
> | Ours         | 0.23     | 0.43     | 26.29 | 25.55 |
>
> It is worth noting that OCE was originally applied to modify cross-attention layers in UNet-based models, whereas FLUX, being a DiT-based architecture, does not have explicit cross-attention. To adapt OCE, we followed the publicly available UCE implementation on FLUX and modified the *context_embedder* and *text_embedder.linear_1* modules instead. Given the structural differences in FLUX, more effective transfer strategies are worth exploring. Additionally, since OCE requires computing an SVD on the matrix $M$, applying it to larger-scale models may incur higher computational costs.
> >**W2 & Q2: More discussions on $C_g$**
>
> For generic concepts $C_g$, we fix them as all tokens from the COCO-30K dataset. We compute the second-order moment of their CLIP text embeddings, resulting in a matrix $K_0$ that serves as a global preservation prior and keeps fixed during the erasure. The computation is efficient (~3s on A100) and performed offline.
>
> To further study the effect of the number of $C_g$ on erasure performance, we conduct an ablation by reducing the size of $C_g$ to **2/3** and **1/3** of the original, as well as evaluating a setting **without $C_g$** in the multi-concept erasure scenario. In the latter case, we replace the original preservation term $||PW C_0 - WC_0||$ with $||PW C_n - WC_n|| + ||PW - W||$, which changes the closed-form solution’s preservation component to $W(I + C_n C_n^T) W^T$. As shown in the table, increasing the number of generic concepts $C_g$ generally improves the balance between erasure ($H_o$) and preservation (FID), confirming the effectiveness of the global prior.
>
> | $C_g$ | $Acc_e$ | $Acc_r$ | $H_o$  | FID(COCO)   | CLIP(COCO)  |
> |:-------------:|:-------:|:-------:|:------:|:-----:|:-----:|
> | Null          | 6.72    | 94.32   | 93.80  | 22.76 | 26.14 |
> | 1/3           | 4.47    | 93.44   | 94.47  | 19.31 | 26.31 |
> | 2/3           | 3.85    | 93.63   | 94.87  | 18.60 | 26.35 |
> | **Ours**          | 3.44   | 94.42   | **95.48**  | **18.33** | 26.33 |
>
> > **Q3: Limitations**
>
> Thank you for the suggestion. We acknowledge that discussing limitations and failure modes is important.
>
> Besides the limitations discussed in Q1 regarding applications on models like FLUX, OCE has some concerns in multi-concept erasure scenarios. Since OCE uses subspace-level constraints rather than strict vector-wise constraints, the erased model may not perfectly generate the intended fine-grained anchor concepts. For example, when erasing 100 celebrities with the anchor concept set as “Taylor Swift,” all 100 celebrities are effectively removed, but the generated face may not strictly correspond to Taylor Swift and instead falls in an intermediate space. This can limit OCE in editing tasks. Additionally, further exploration is needed for erasure of more implicit concepts such as relational, compositional understanding, or watermarks.

---

> > ### Author Rebuttal · Reviewer_kex6 · 2026-04-01
> >
> > Thank you for your detailed rebuttal. The authors’ responses satisfactorily address my questions. I support the acceptance of this paper and will maintain my initial score.

---

> > > ### Author Response · Authors · 2026-04-02
> > >
> > > Thank you for the kind feedback and support. We're glad our responses addressed your questions, and we truly appreciate your thoughtful comments.

---

### Official Review · Reviewer_Nk6C · 2026-03-12

**Soundness:** 3
**Presentation:** 3
**Significance:** 3
**Originality:** 3
**Overall Recommendation:** 4
**Confidence:** 4

**Summary:**

This paper proposes Orthogonal Concept Erasure (OCE), which replaces the additive parameter updates used by existing editing-based erasure methods with multiplicative orthogonal transformations. The motivation is geometric: concept semantics depend on neuron direction, while generation quality depends on inter-neuron angular geometry. Orthogonal transforms can rotate directions while exactly preserving magnitudes and angles, avoiding the entanglement inherent in additive updates. Experiments on SD v1.4 and FLUX across four erasure tasks show OCE outperforms 7 baselines, erasing up to 100 concepts in efficiently,

**Compliance With Llm Reviewing Policy:**

Affirmed.

**Key Questions For Authors:**

* While the concept erasure literature is large and it is difficult to compare against every method, given that multi-concept erasure is one of the key strengths claimed by OCE, a comparison with CURE is necessary.

**Limitations:**

* Can the authors provide some ablation on the hyperparameters $\lambda_c, \lambda_c, \lambda_0, \lambda_r$? How are these hyperparameters tuned?

**Strengths And Weaknesses:**

* The paper is well-written and the problem is well-motivated.
* The reformulation as an orthogonal Procrustes problem with a closed-form SVD solution is elegant and avoids iterative optimization.
* Experimental results are comprehensive,  spanning four erasure tasks (object, style, celebrity, NSFW), two model architectures (SD v1.4, FLUX), and single- and multi-concept settings.
* The scalability result (100 concepts in 4.3 seconds) is a strong practical advantage.

---

> ### Author Rebuttal · Authors · 2026-03-31
>
> >**Q1: Comparison with CURE**
>
> Thank you for the suggestion. Unfortunately, CURE’s GitHub repository does not release the training code and only provides three checkpoints—“cat” (object erasure), “Kelly Mckernan” (style erasure), and “Angelina Jolie” (celebrity erasure). Therefore, we compare our OCE with CURE using these available checkpoints:
>
> *Erasing "cat"*
> |Method|$Acc_e$|$Acc_r$|$H_o$|
> |:-:|:-:|:-:|:-:|
> |SD v1.4|98.93|98.60|2.12|
> |CURE|8.84|95.02|93.05|
> |Ours|0.15|98.71|**99.28**|
>
> *Erasing "Kelly Mckernan"*
> |Method|CS|FID(COCO)|CS(COCO)|
> |:-:|:-:|:-:|:-:|
> |SD v1.4|27.60|-|26.56|
> |CURE|**14.76**|13.11|26.55|
> |Ours|15.52|**11.10**|26.55|
>
> *Erasing "Angelina Jolie"*
> |Method|$Acc_e$|$Acc_r$|$H_o$|FID(COCO)|CS(COCO)|
> |:-:|:-:|:-:|:-:|:-:|:-:|
> |SD v1.4|93.31|97.51|12.52|-|26.56|
> |CURE|27.36|83.84|77.84|22.75|**26.87**|
> |Ours|0.00|97.74|**98.86**|**7.15**|26.58|
>
> Overall, we observe that OCE consistently achieves stronger concept erasure while effectively preserving unrelated content compared with CURE.
> >**L1: Ablation study on the hyperparameters**
>
> Thank you for the suggestion. We conduct an ablation study on $\lambda_e$, $\lambda_0$, and $\lambda_r$ in the multi-concept erasure scenario. The results are summarized in the table below. From the table, we can observe:
> - $\lambda_e$ mainly affects the **erasure effectiveness** ($Acc_e$)
> - $\lambda_0$ mainly affects the **overall generation quality** (FID(COCO))
> - $\lambda_r$ mainly affects the **preservation of neighboring concepts** ($Acc_r$).
>
> These findings are consistent with our design. In our experiments, we choose **$\lambda_e=900$, $\lambda_0=50$, $\lambda_r=3$** to achieve a balanced trade-off between erasure and preservation.
>
> |$\lambda_e$|$\lambda_0$|$\lambda_r$|$Acc_e$|$Acc_r$|$H_o$|FID(COCO)|CLIP(COCO)|
> |:-:|:-:|:-:|:-:|:-:|:-:|:-:|:-:|
> |**900**|**50**|**3**|3.44|94.42|**95.48**|18.33|26.33|
> |1000|50|3|2.85|93.25|95.16|19.01|26.29|
> |1200|50|3|2.03|89.73|93.67|19.69|26.24|
> |600|50|3|7.70|93.48|92.89|17.61|26.41|
> |900|20|3|3.05|94.01|95.46|21.05|26.12|
> |900|80|3|5.67|93.85|94.09|17.55|26.44|
> |900|50|1|1.84|89.57|93.67|18.63|26.34|
> |900|50|5|6.52|94.49|93.98|18.34|26.37|

---

> > ### Author Rebuttal · Reviewer_Nk6C · 2026-04-05
> >
> > Thank you for the additional experiments. My concerns are addressed. Hence, I would like to keep my score.

---

> > > ### Author Response · Authors · 2026-04-06
> > >
> > > Thank you for your positive feedback and support. We're glad that our additional experiments addressed your concerns, and we truly appreciate your helpful comments.

---

### Official Review · Reviewer_PuRd · 2026-03-13

**Soundness:** 3
**Presentation:** 3
**Significance:** 3
**Originality:** 3
**Overall Recommendation:** 5
**Confidence:** 4

**Summary:**

The paper proposes orthogonal concept erasure (OCE), which is a method to unlearn unwanted concepts from text-to-image diffusion models. The method is based on the insight that the angles between neuron directions is much more important for preserving generation quality than previously assumed. Therefore, OCE does not change the neuron magnitudes, and more importantly, the angles between the neuron directions. In the experimental evaluation the method performs better than the baseline on average, while preserving the generation quality.

**Compliance With Llm Reviewing Policy:**

Affirmed.

**Final Justification:**

All my concerns have been addressed, and I retain my score of accept.

**Key Questions For Authors:**

Q1: How is the retain set split into generic concepts and neighboring concepts? Is this done by hand? The generic concepts are pre-computed, but how is it decided what the neighboring concepts are?
Q2: How are the anchor concepts selected?
Q3: Is there any explanation for why, in Tab. 1, OCE performs so much worse on some of the concepts than the baseline methods?

**Limitations:**

Yes

**Strengths And Weaknesses:**

Strengths:
- The observation that the angles between the neuron directions are important for concept erasure seems very novel
- The idea has a very strong theoretical foundation
- The OCE method outperforms existing methods
- The proposed method is more efficient than previous methods

Weaknesses:
- It seems that there is a high difference in the success of erasing different concepts (see Tab. 1).


Misc:
- Line 59, right column: Typo in "Orthigonal"
- Why is INPO listed as a concurrent ICML submission? I can see that the paper was rejected from ICLR. But how could the authors know that this paper was submitted to ICML?

---

> ### Author Rebuttal · Authors · 2026-03-31
>
> > **W1: There is a high difference in the success of erasing different concepts**
>
> Thank you for the question. We first clarify that this phenomenon is consistently observed across all baselines. As shown in Tab. 1, categories such as “airplane” and “automobile” tend to have higher $Acc_e$. We attribute this difference primarily to the **evaluation metric**. Specifically, following prior works, we adopt a CLIP score + softmax classifier over CIFAR-10 classes. All baselines are evaluated under the same criterion for fair comparison. However, we observe that this metric introduces category-dependent bias due to its closed-set classification nature. For example, after erasing the concept “airplane”, the generated image may become a clean sky without airplane. Nevertheless, the classifier may still assign a relatively high probability to the original class. A similar effect occurs for “automobile”, where background cues (e.g., roads or urban backgrounds) still trigger the classifier. Therefore, the higher $Acc_e$ does not necessarily indicate incomplete erasure, but also reflects the characteristics of the metric.
>
> > **Misc: Typo and Concurrent Submission**
>
> Thank you for pointing this typo. We will carefully revise the manuscript.
>
> Regarding concurrent submission, we strictly follow the ICML policy on dual and concurrent submissions:
> > *In particular, concurrent submissions to ICML with an overlapping set of authors are considered prior work. If a concurrent submission is on a related research topic that would be normally included in the related work section of the paper, it must be discussed as related work and cited in the body of the paper (an anonymized PDF of the concurrent submission must be provided in the supplementary material).*
>
> > **Q1: Selection of generic and neighboring concepts**
>
> Thank you for the question. **For generic concepts**, we fix them as all tokens from the COCO-30K dataset. We compute the second-order moment of their CLIP text embeddings, resulting in a matrix $K_0$ that serves as a global preservation prior and keeps fixed during the erasure. The computation is efficient (~3s on A100) and performed offline. **For neighboring concepts**, we simply use GPT-5 to generate semantically related concepts, which capture the local semantic neighborhood of the target concept. Exploring more principled neighboring concept selection is an interesting direction for future work.
> > **Q2: Selection of anchor concepts**
>
> Thank you for the question. For anchor selection, we adopt a heuristic strategy: anchor concepts are chosen to belong to the same high-level category as the target concept, sharing some similarity but also exhibiting noticeable differences. This ensures smooth and stable concept erasure. To further investigate the impact of anchor choice, we conducted experiments on CIFAR-10 and multi-concept erasure scenarios. We compared our heuristic selections with **random anchors** (in CIFAR-10) and **empty anchor**:
>
> *CIFAR-10*
> |Anchor||Airplane|||Automobile|||Bird|||Cat|||Deer||
> |-|:-:|:-:|:-:|:-:|:-:|:-:|:-:|:-:|:-:|:-:|:-:|:-:|:-:|:-:|:-:|
> ||$Acc_e$|$Acc_r$|$H_o$| $Acc_e$|$Acc_r$|$H_o$|$Acc_e$|$Acc_r$|$H_o$|$Acc_e$|$Acc_r$|$H_o$|$Acc_e$|$Acc_r$| $H_o$|
> |empty|9.22| 97.63| 94.08|12.61|97.13| 92.00|3.12|97.90|97.38|5.03|98.41|96.66|6.90|98.47|95.71|
> |random|7.71|98.62|95.35|10.91|98.56|93.59|5.62|97.36|95.85|2.49|98.55|98.02|4.28|98.32|97.00|
> |Ours|6.89|98.85|**95.89**|8.79|99.07|**94.98**|3.40|98.30|**97.44**|0.15|98.71|**99.28**|3.81|98.45|**97.31**|
>
> *Multi-concept (100 Celebs)*
> |Anchor|$Acc_e$|$Acc_r$|$H_o$|FID(COCO)|CS(COCO)|
> |:-:|:-:|:-:|:-:|:-:|:-:|
> |empty|4.28|91.58|93.60|18.64|26.04|
> |tree|4.48|94.68|95.10|18.18|25.85|
> |cat|4.24|92.20|93.95|18.45|25.92|
> |person|3.85|94.50|95.32|18.29|26.40|
> |Ours|3.44|94.42|**95.48**|18.33|26.33|
>
> Overall, the performance varies under different anchor settings, and our heuristic strategy consistently improves erasure effectiveness.
> >**Q3: Why OCE performs worse on some of the concepts**
>
> Thank you for the question. In Tab.1, OCE indeed achieves lower erasure performance than UCE on "automobile" concept. However, as discussed in Q1, this is closely related to the characteristics of the evaluation metric. OCE is designed as a relatively conservative erasure strategy, which tends to preserve more background and contextual information. For "automobile", after erasure, some generated images may be only urban or road-like backgrounds. Such images can still be mistakenly classified as "automobile" under     the closed-set classifier. In contrast, UCE performs more aggressive, vector-wise alignment toward anchors, leading to larger distributional shifts in the generated images. This results in lower ACC, but also produces more drastic visual changes. Therefore, the observed difference in this metric mainly reflects the different erasure behaviors. We will include qualitative comparisons in the revision to better illustrate this difference.

---

> > ### Author Rebuttal · Reviewer_PuRd · 2026-04-01
> >
> > Thank you for the clarifications. All my concerns have been addressed.

---

> > > ### Author Response · Authors · 2026-04-02
> > >
> > > Thank you for your positive feedback and support. We're glad that our clarifications helped address your concerns, and we truly appreciate your helpful comments.

---

### Decision · Program_Chairs · 2026-04-30

**Decision:**

Accept (spotlight)

**Comment:**

The paper proposes a novel method for concept erasure that relies on the angles between neuron directions rather than the magnitudes and uses this for closed-form editing.

The reviewers agree that the paper tackles a timely topic, that the method is novel and provides strong theoretical perspective on the topic of concept editing.

The rebuttal provided a significant body of experiments to address all the concerns raised by the reviewers. Especially the addition of the experiments on FLUX and the robustness evaluation will be a valuable add on for the final version of the paper.